# Non-invasive monitoring of alternative splicing outcomes to identify candidate therapies for myotonic dystrophy type 1

Ningyan Hu[1,2], Layal Antoury[1,2], Timothy M. Baran[3], Soumya Mitra[3], C. Frank Bennett[4], Frank Rigo[4], Thomas H. Foster[3] & Thurman M. Wheeler[1,2]

During drug development, tissue samples serve as indicators of disease activity and phar-macodynamic responses. Reliable non-invasive measures of drug target engagement will facilitate identification of promising new treatments. Here we develop and validate a novel bi-transgenic mouse model of myotonic dystrophy type 1 (DM1) in which expression of either DsRed or GFP is determined by alternative splicing of an upstream minigene that is mis-regulated in DM1. Using a novel in vivo fluorescence spectroscopy system, we show that quantitation of the DsRed/GFP ratio provides an accurate estimation of splicing outcomes in muscle tissue of live mice that nearly doubles throughput over conventional fluorescence imaging techniques. Serial in vivo spectroscopy measurements in mice treated with a C16 fatty acid ligand conjugated antisense (LICA) oligonucleotide reveal a dose-dependent therapeutic response within seven days, confirm a several-week duration of action, and demonstrate a two-fold greater target engagement as compared to the unconjugated parent oligonucleotide.

[1] Department of Neurology, Massachusetts General Hospital, Boston, MA 02114, USA. [2] Harvard Medical School, Boston, MA 02114, USA. [3] Department of Imaging Sciences, University of Rochester, Rochester, NY 14642, USA. [4] Ionis Pharmaceuticals, Carlsbad, CA 92010, USA. Correspondence and requests for materials should be addressed to T.M.W. (email: twheeler1@mgh.harvard.edu)

Muscular dystrophies are a heterogeneous group of genetic disorders that cause progressive muscle weakness and wasting. New therapies for muscular dystrophies, and better assays to find these therapies, are critical unmet needs. An obstacle to developing new treatments for muscular dystrophies has been the lack of appropriate animal models that enable assays for efficient screening of new candidate drugs. Current methods of measuring drug activity involve biochemical analysis of muscle tissue, which is expensive, time consuming, and may require large numbers of animals to achieve the necessary statistical power. Development of new model systems that enable rapid in vivo detection of pharmacodynamic activity is essential to speed drug discovery.

Myotonic dystrophy (dystrophia myotonica; DM) is the most common muscular dystrophy in adults, affecting ~1 in 7500[1]. The genetic cause of DM type 1 (DM1) is a CTG repeat expansion (CTG$^{exp}$) in the 3′ untranslated region of the DM protein kinase (DMPK) gene. Expression of mutant DMPK-CUG$^{exp}$ mRNA in muscle results in myotonia (delayed relaxation of muscle fiber contraction), histopathologic myopathy, and progressive muscle wasting. No current treatment alters the disease course. Symptoms in DM1 arise from a novel RNA-mediated disease mechanism that involves the inhibition of alternative splicing regulator proteins by mutant DMPK-CUG$^{exp}$ transcripts, resulting in inappropriate expression of developmental splice isoforms in adult tissues[2–4]. In DM1 patients, RNA splice variants serve as biomarkers of disease activity[5], while in DM1 mice they also serve as sensitive indicators of therapeutic drug response[6–8], which can be translated to clinical care.

The discovery and development of genetically encoded fluorescent proteins has enabled multicolor imaging of biological processes such as differential gene expression and protein localization in living cells[9,10]. Genetic modifications of green fluorescent protein (GFP), derived from the jellyfish Aequorea victoria, and red fluorescent protein DsRed, derived from the coral Discosoma species, have improved the brightness and photostability of these proteins. A previous study capitalized on an unusual feature of the DsRed gene, which is that it has two open reading frames, to demonstrate that GFP and DsRed can be used together in a single bi-chromatic construct to quantify alternative splicing events within individual cells or mixed cell populations using flow cytometry or fluorescence microscopy[11].

Here we modify and optimize this bi-chromatic reporter for in vivo use, generate a novel therapy reporter (TR) transgenic mouse model of DM1 that expresses the optimized construct, design and custom-build an in vivo fluorescence spectroscopy system for rapid measurements of splicing outcomes in live TR mice, and test the pharmacodynamic activity of a novel ligand conjugated antisense (LICA) oligonucleotide that is designed to enhance drug uptake into target tissues as compared to the unconjugated parent ASO.

## Results

**Bi-chromatic alternative splicing therapy reporter.** In DM1, a hallmark of cellular dysfunction is inappropriate expression of developmental splice isoforms in adult tissues. The Human Skeletal Actin - Long Repeat (HSA$^{LR}$) transgenic mouse model[3] was designed to test the hypothesis that expanded CUG repeat mRNA is toxic for muscle cells. In this model, human skeletal actin (ACTA1) transcripts that contain ~220 CUG repeats are expressed in muscle tissue, resulting in myotonia (delayed muscle relaxation due to repetitive action potentials) and histopathologic signs of myopathy[3]. Muscle tissue of HSA$^{LR}$ mice features misregulated alternative splicing that is highly concordant with

human DM1 muscle tissue, including preferential exclusion of Atp2a1 exon 22[2,5].

A previously published fluorescence bi-chromatic reporter construct[11] used a ubiquitous CMV promoter to mediate activity of the construct, and splicing of a chicken cardiac troponin T (Tnn2) exon 5 minigene to determine the reading frame. Only one of these reading frames encodes DsRed. By placing the DsRed cDNA adjacent to the GFP cDNA, two mutually exclusive reading frames resulted in production of either DsRed when exon 5 was included, or GFP when exon 5 was excluded. The GFP reading frame has a long N-terminal peptide encoded by the alternate open reading frame of DsRed, but fluorescence of GFP remained bright. To determine whether this construct is viable as an in vivo reporter of alternative splicing derangements in DM1 muscle tissue, we injected and electroporated plasmid DNA[12] that contains the construct into tibialis anterior (TA) muscles in HSA$^{LR}$ mice and WT controls. In vivo DsRed/GFP fluorescence of injected muscles appeared similar in HSA$^{LR}$ and WT mice, with exon 5 exclusion predominating in both (Supplementary Fig. 1).

To optimize the bi-chromatic construct for in vivo expression in skeletal muscle and enable detection of differential splicing in DM1 muscle, we modified the construct in two ways. First, we replaced the chicken Tnnt2 minigene with a human ATP2A1 exon 22 minigene. Of the dozens of transcripts mis-spliced in DM1 muscle, ATP2A1 exon 22 was chosen for this reporter because it has the largest change in DM1 mouse muscle that we have observed, and is highly responsive to therapeutic ASOs, as measured by traditional RT-PCR analysis of muscle tissue samples[7,8]. To bypass a stop codon and enable shift of the reading frame[13], exon 22 of the ATP2A1 minigene was modified by site-directed mutagenesis to induce a single base pair deletion so that it contains 41 base pairs instead of 42. Second, to restrict expression to skeletal muscle, we replaced the CMV promoter with a human skeletal actin (HSA) promoter[14]. In this system, inclusion of APT2A1 minigene exon 22 (high in WT and ASO-treated DM1 mouse muscle) gives rise to a transcript that produces a red fluorescent protein (DsRed), while exclusion of exon 22 (high during muscle regeneration and in adult DM1 muscle) shifts the reading frame from DsRed to GFP, resulting in a transcript that produces green fluorescent protein (Fig. 1a). By design, measurement of the DsRed/GFP ratio by quantitative imaging enables determination of therapeutic response in DM1 mice. Intramuscular injection of HSA-therapy reporter (TR) plasmid DNA was associated with a significantly higher DsRed/GFP ratio by in vivo imaging in WT mice than in HSA$^{LR}$ transgenic mice (Supplementary Fig. 2). RT-PCR analysis of skeletal muscle tissue RNA also identified a higher percent inclusion of exon 22 inclusion in WT muscle than HSA$^{LR}$ muscle, confirming that the TR construct splice switch is active in vivo (Supplementary Fig. 2). In cryosections of injected muscles, DsRed expression was higher in WT than in HSA$^{LR}$ muscle and localized to the cytoplasm, while GFP expression was higher in HSA$^{LR}$ than in WT and, unexpectedly, localized to myonuclei instead of cytoplasm (Supplementary Fig. 2).

**Novel TR transgenic mouse model.** Transgenic mice were generated by microinjection of a linear HSA-TR construct DNA (see Methods). Expression of the TR transgene was high by RT-PCR in all skeletal muscles examined, and low or absent in heart and liver, while exon 22 splicing pattern of the TR-ATP2A1 transgene (Tg) appeared similar to the exon 22 splicing pattern of endogenous mouse Atp2a1 in all muscles examined except the flexor digitorum brevis (Supplementary Fig. 3). Wild-type TA muscle injected with TR plasmid (TR-P) served as a control.

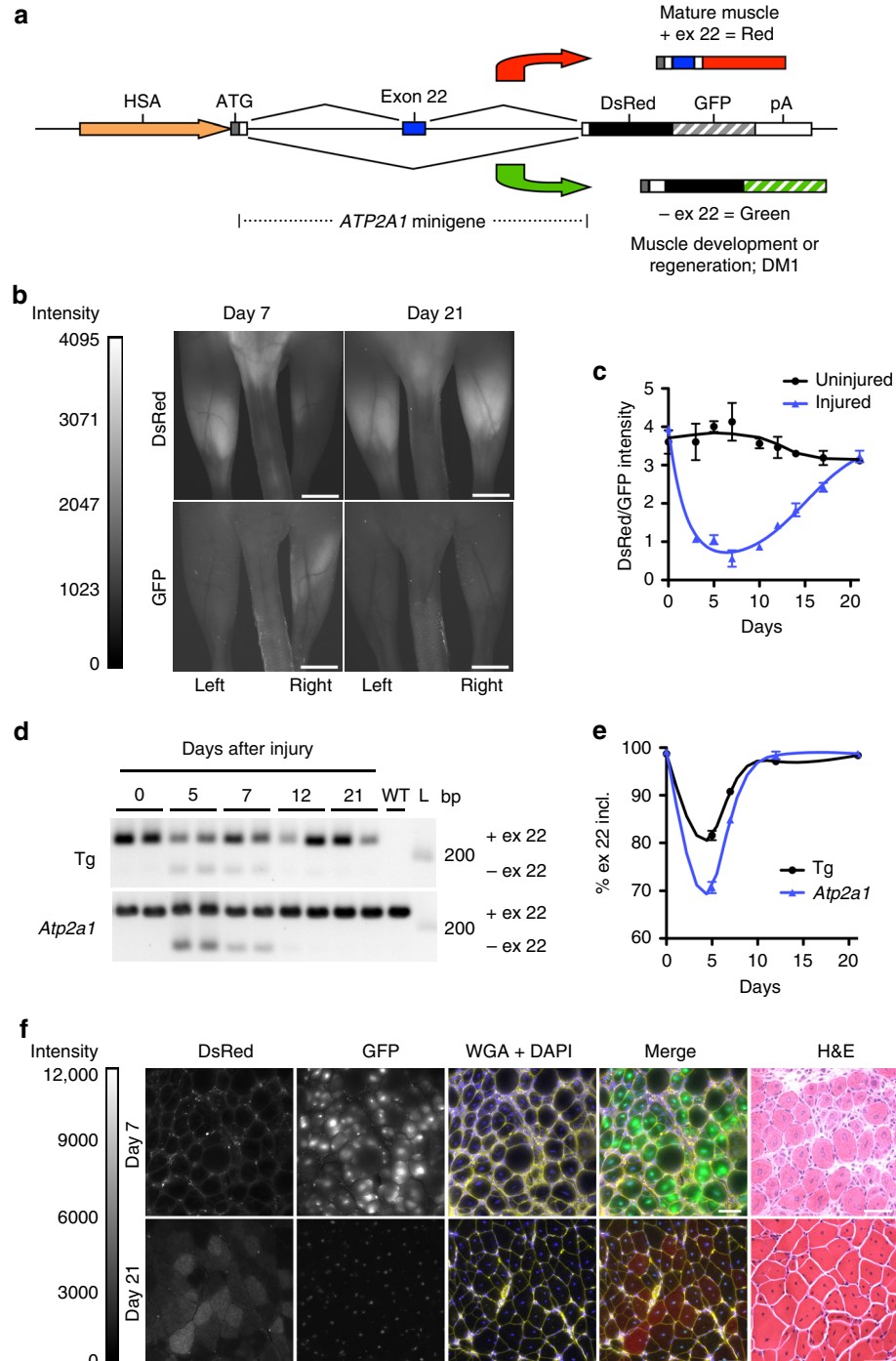

**Fig. 1** Design and validation of the therapy reporter (TR) splicing construct. **a** TR construct design. **b** To test the splice switch under conditions of muscle regeneration, we induced acute injury by injecting 1.2% barium chloride[17] into the right gastrocnemius muscles of TR transgenic mice ($N = 8$) and monitored quantitative DsRed/GFP ratios by serial fluorescence microscopy of live mice. The left gastrocnemius was untreated. Shown are representative images of DsRed and GFP fluorescence on Days 7 and 21 after acute injury. Mice are prone. Intensity range = 0–4095 grayscale units. Bars = 4 mm. **c** Quantitative DsRed/GFP fluorescence ratios in untreated (black circles) and injured (blue triangles) gastrocnemius muscles by serial in vivo imaging after acute injury. Error bars indicate mean ± s.e.m. Non-linear regression. The imaging results are representative of four independent experiments. **d** Alternative splicing analysis by RT-PCR of transgene (Tg) and endogenous (Endo) *Atp2a1* exon 22 on 0, 5, 7, 12, and 21 days after injury ($N = 2$ mice each time point). WT wild-type control, L DNA ladder, bp base pairs. **e** Quantitation of splicing shown in **d**, displayed as % exon 22 inclusion. Error bars indicate mean ± s.e. m. Non-linear regression. **f** DsRed and GFP fluorescence in gastrocnemius muscle tissue sections 7 and 21 days after acute injury. Fluorescence images are the extended focus of deconvolved Z series. Fluorescence intensity range = 0–12,000 grayscale units. Alexa 647-wheat germ agglutinin (WGA; pseudocolored yellow) and DAPI (blue) highlight muscle fibers and nuclei, respectively. Merge DsRed (red) + GFP (green) + WGA + DAPI, H&E hematoxylin and eosin. Bars = 50 μm

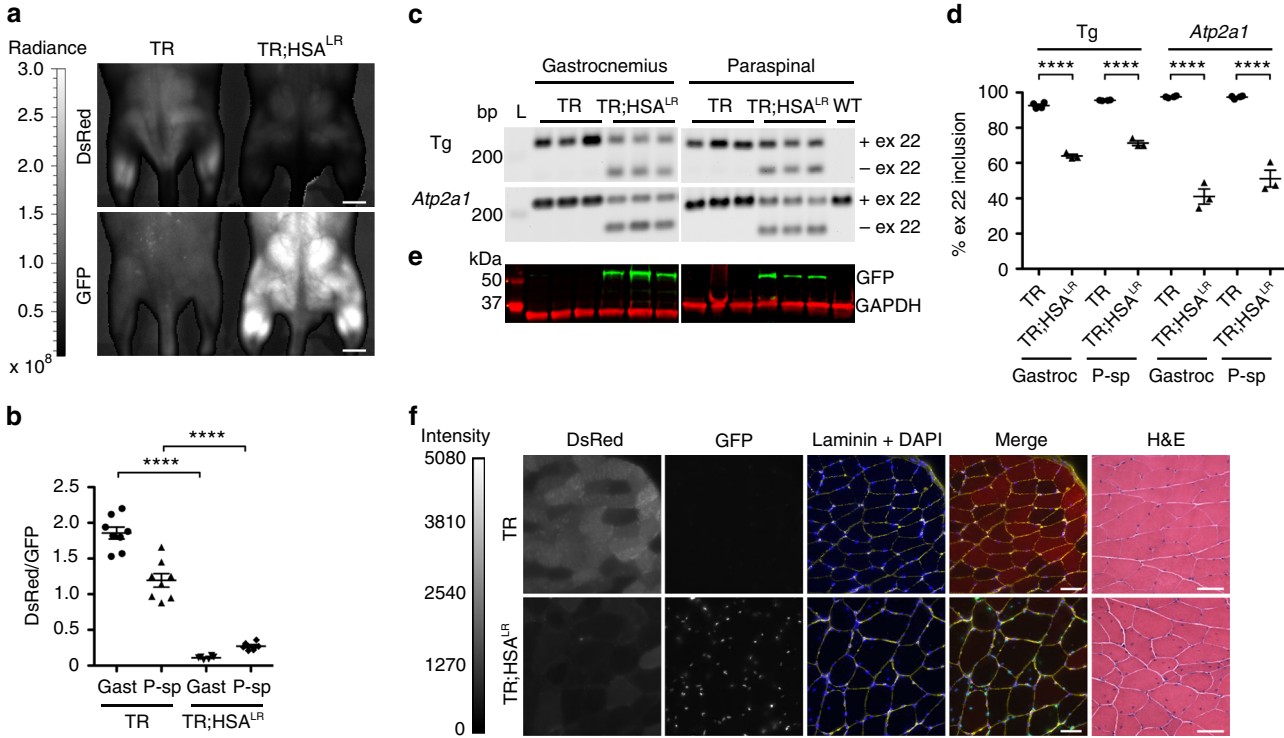

**Fig. 2** TR;HSA[LR] bi-transgenic mice as a therapy reporter model for DM1. In the HSA[LR] transgenic mouse model of DM1, exclusion of *Atp2a1* exon 22 is upregulated. To validate the TR-*ATP2A1* splice switch as a reporter for DM1, we crossed the TR transgenic with the *HSA*[LR] transgenic model to create TR; *HSA*[LR] bi-transgenic mice. **a** In vivo imaging of DsRed and GFP fluorescence in TR transgenic and TR;*HSA*[LR] bi-transgenic mice (IVIS Spectrum) under general anesthesia in the prone position. The scale shows radiant efficiency (photons/second/cm2). Bars = 5 mm. **b** DsRed and GFP fluorescence measured in regions of interest in bilateral gastrocnemius (Gast) and paraspinal (P-sp) muscles of TR ($N = 4$) and TR;HSA[LR] mice ($N = 3$). Error bars indicate mean ± s.e.m. ****$P < 0.0001$ TR vs. TR;*HSA*[LR] (one-way ANOVA). **c** Splicing analysis in gastrocnemius muscle by RT-PCR of the TR transgene (Tg) exon 22 (ex 22) and endogenous *Atp2a1* ex 22. $N = 3$ each group. WT FVB wild-type. **d** Quantitation of splicing results in **c**, displayed as the percentage of exon 22 inclusion. Error bars indicate mean ± s.e.m. ****$P < 0.0001$ (two-way ANOVA). **e** Western blot of GFP protein expression in gastrocnemius and paraspinal muscles of TR and TR;HSA[LR] mice ($N = 3$ each group), with GAPDH as loading control. kD kilodaltons. **f** Representative DsRed and GFP expression in TR (upper row) and TR;HSA[LR] (lower row) gastrocnemius muscle cryosections. Fluorescence images are the extended focus of deconvolved Z-series. Fluorescence intensity range = 0–5080 grayscale units. Laminin (yellow) and DAPI (blue) highlight muscle fibers and nuclei, respectively. Merge DsRed (red) + GFP (green) + laminin + DAPI, H&E hemotoxylin and eosin. Bars = 50 μm

**Fluorescence imaging of muscle regeneration**. After muscle injury, muscle stem cells, termed satellite cells, proliferate and form muscle precursor cells that differentiate and eventually form new mature muscle fibers. During the differentiation and maturation process, splicing of several mRNAs transitions from developmental to fully mature adult isoforms[15]. Alternative splicing of *ATP2A1* exon 22 is developmentally regulated, being preferentially excluded during muscle development or regeneration of adult muscle, and preferentially included in normal adult muscle[2,16]. To determine whether exon 22 in the TR transgene *ATP2A1* also is developmentally regulated, and to monitor muscle regeneration, we injected the right gastrocnemius (gastroc) of homozygous TR transgenic mice (Supplementary Fig. 3; Supplementary Table 1) with barium chloride (BaCl2) to induce muscle injury[17], and measured fluorescence by serial in vivo imaging under general anesthesia. Quantitative DsRed/GFP measurements in treated muscles were lowest at Day 7 and returned to baseline by Day 21 (Fig. 1b, c; Supplementary Fig. 4; Supplementary Table 2). Splicing by RT-PCR revealed that the peak exon 22 exclusion of both transgene *ATP2A1* and mouse *Atp2a1* was Day 5 after injury, with a return to the baseline splicing pattern by Day 12, similar to a previous report of mouse *Atp2a1* splicing after muscle injury by cardiotoxin[16], and indicating that the splicing recovery, as measured by DsRed/GFP

imaging, is delayed over actual splicing outcomes (Fig. 1c–e). Quantitative imaging of gastrocnemius muscle cryosections revealed high GFP expression at Day 7, during active muscle regeneration, and low GFP expression by Day 21, when formation of new muscle fibers is largely complete (Fig. 1f). DsRed fluorescence was concentrated in muscle fiber cytoplasm, while GFP fluorescence was confined to myonuclei, similar to the localization after intramuscular injections of plasmid DNA (Fig. 1f).

**Bi-transgenic mouse model of DM1**. To test the TR construct as a splicing biomarker for DM1, we crossed TR transgenic mice with HSA[LR] transgenic mice (both on the FVB background) to produce homozygous TR[+/+];HSA[LR+/+] bi-transgenic mice (Supplementary Fig. 3; Supplementary Table 1). Quantitative in vivo fluorescence imaging of gastrocnemius and lumbar paraspinal muscles of TR;HSA[LR] bi-transgenic mice revealed a low DsRed/GFP ratio, as compared to TR single transgenic mice (Fig. 2a, b). RT-PCR analysis of muscle tissue confirmed the splice switch in TR;HSA[LR] mice, while higher GFP expression also was evident in bi-transgenic mice by Western blot (Fig. 2c–e). Quantitative fluorescence microscopy of muscle cryosections demonstrated bright DsRed and low GFP expression in TR single transgenic, and low DsRed and high GFP expression in TR;HSA[LR] bi-transgenic mice (Fig. 2f). Muscle histology in TR

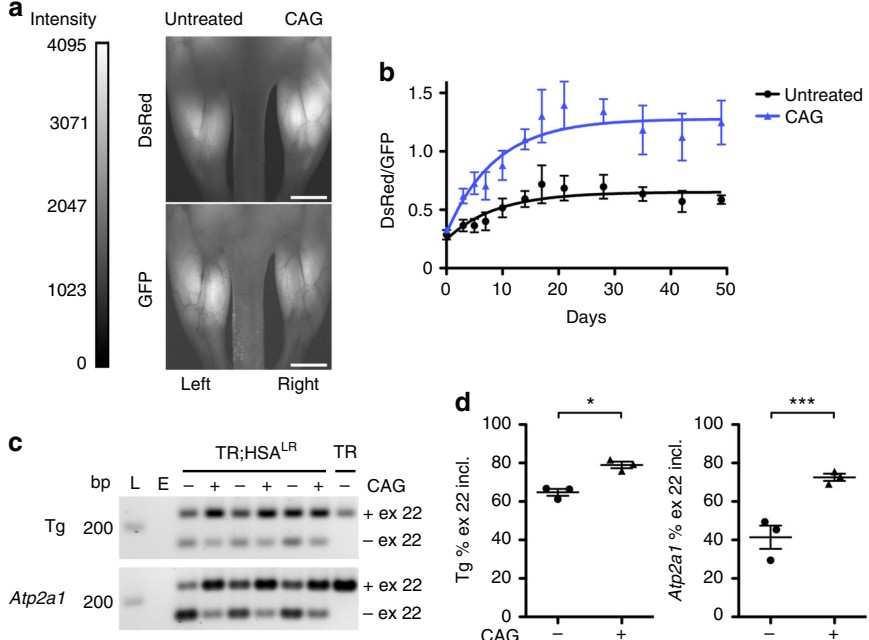

**Fig. 3** Imaging therapeutic antisense oligonucleotide drug activity in vivo. We administered a CAG repeat morpholino (CAG) ASO designed to bind and neutralize the pathogenic effects of expanded CUG repeat RNA[7] by intramuscular injection of the right gastrocnemius in TR;HSA[LR] mice ($N = 3$). The left gastrocnemius was injected with saline or was untreated. **a** Representative images of quantitative in vivo DsRed and GFP fluorescence under general anesthesia 28 days after injection of the CAG ASO. Fluorescence intensity range = 0–4095 grayscale units. Bars = 4 mm. **b** Quantitative DsRed/GFP fluorescence ratios in muscles of untreated (black circles) and CAG ASO-treated (blue triangles) mice by serial in vivo imaging. Error bars indicate mean ± s.e.m. Non-linear regression. **c** RT-PCR analysis of transgene and endogenous *Atp2a1* exon 22 in untreated (−) or CAG ASO-treated ( + ) muscles 49 days after injection. E empty lane, L DNA ladder, bp base pairs. **d** Quantitation of splicing results in **c**. Error bars indicate mean ± s.e.m. ***$P = 0.0002$; *$P$ = mean difference 14.2, 95% CI of difference 1.0–27.5 (two-way ANOVA). These results are representative of four independent experiments

transgenic mice was similar to wild-type, and TR;HSA[LR] bi-transgenic mice appeared similar to *HSA*[LR] single transgenic mice (Fig. 2f).

**TR construct as a pharmacodynamic biomarker in DM1 mice.** Antisense oligonucleotides (ASOs) are modified nucleic acid molecules that bind to RNA through Watson–Crick bonds. The chemistry and binding target of each ASO determine therapeutic effects[18]. In DM1 transgenic mice, ASOs that target the RNA-mediated disease process can reverse RNA mis-splicing, eliminate myotonia, and slow myopathy progression by reducing the pathogenic effects of expanded CUG repeat (CUG[exp]) RNA[7,8]. One approach for treatment of DM1 involves ASOs or small molecules that are designed to bind directly to the CUG[exp] RNA and reduce pathogenic interactions with MBNL proteins by steric inhibition[7,19]. To validate the TR construct as an indicator of therapeutic ASO activity, we injected gastrocnemius muscles of TR;HSA[LR] bi-transgenic mice with saline or 20 μg of a dendrimer-modified CAG repeat morpholino ASO (CAG25)[7] (Fig. 3). Serial in vivo imaging demonstrated an increase of the DsRed/GFP fluorescence ratio in ASO-treated muscles by Day 3 after injection, which persisted for at least several weeks after injection (Fig. 3a, b). Splicing analysis by RT-PCR of muscle tissue harvested at Day 49 confirmed correction of the splicing in muscles treated with ASOs (Fig. 3c, d).

**In vivo spectroscopy to monitor splicing correction.** In vivo molecular imaging using a fluorescence microscope or an IVIS system allows rapid visualization of heterogeneous gene expression in muscle tissue. Laser-based spectroscopy using dedicated optical fibers for excitation and emission is an alternative

approach for in vivo fluorescence measurements that may offer a more sensitive ratiometric analysis than imaging techniques, due in part to more accurate separation of overlapping emission spectra, such as GFP and DsRed, in the presence of an auto-fluorescence background that is typical of biological tissues. The use of a dedicated spectroscopy system with laser sources for excitation also allows for the detection of relatively small quantities of the fluorophores being expressed and more accurate tracking of disease progression or regression, even at very early and late stages.

A laser-excitation-based fluorescence spectroscopy system was constructed that is similar to the general design of previously reported spectroscopy instruments[20–22]. For in vivo detection of emission spectra, the system includes a 488 nm laser (GFP), a 532 nm laser (DsRed), separate optical fibers for excitation and emission detection, and a portable spectrometer (Fig. 4a, b; Supplementary Table 3). A broadband white light source enables white light spectroscopy for correction of detected fluorescence spectra for any effects of varying tissue optical properties.[23] This correction is vital, as it ensures that changes in detected fluorescence are due only to changes in fluorophores rather than the effects of changes in the intervening muscle tissue due to ASO or other drug treatments. Spectral analysis using singular value decomposition (SVD) isolates the GFP and DsRed spectra and removes autofluorescence[24–26], thereby enabling a rigorous estimate of the relative amplitudes of GFP and DsRed. The entire spectroscopy system is enclosed and controlled via LabVIEW software.

To test and calibrate the spectroscopy system for in vivo measurements, we injected TA muscles of wild-type mice with GFP or DsRed plasmid DNA driven by the same muscle-specific

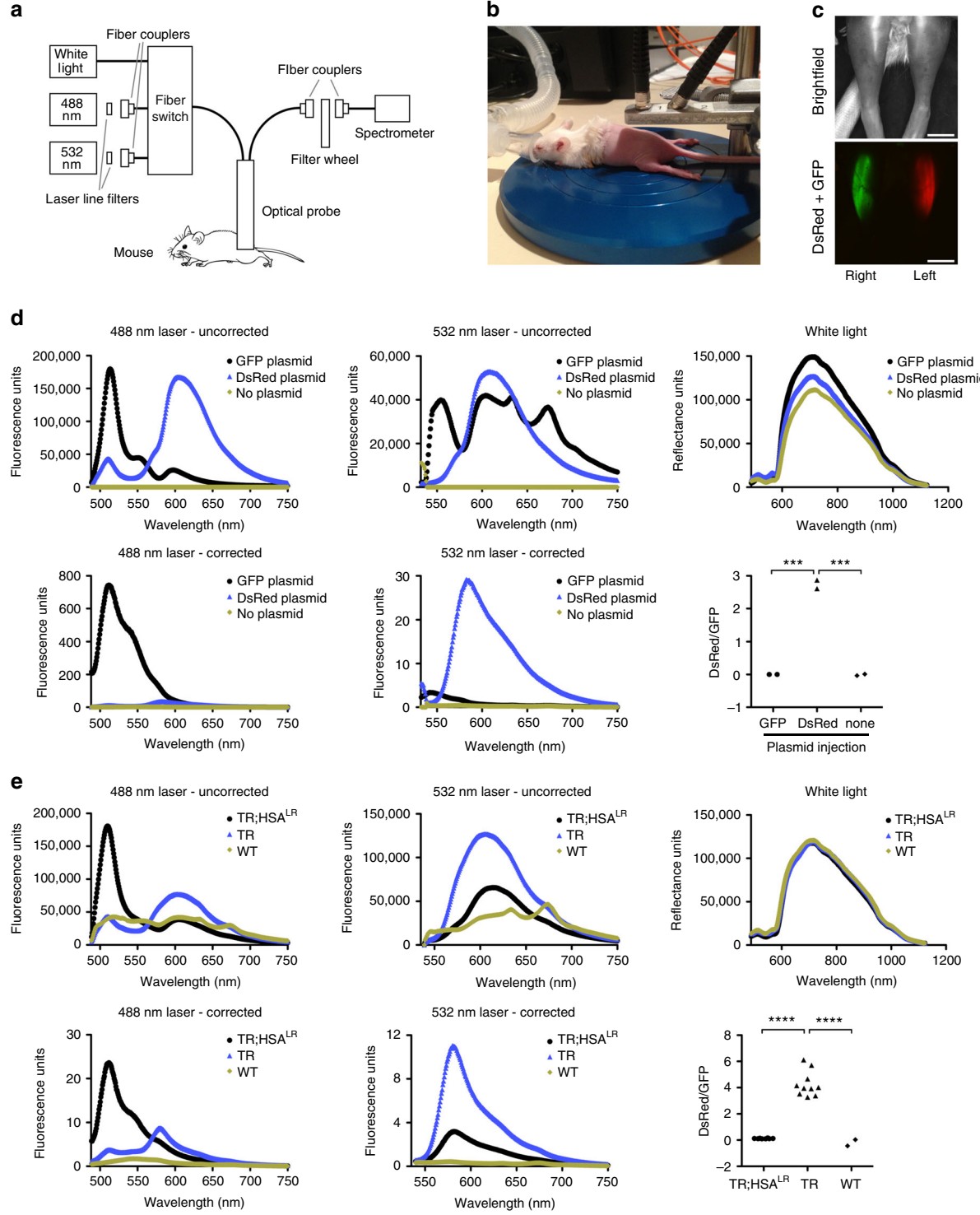

HSA promoter[14] that was used for expression of the TR transgene (Fig. 4c; Supplementary Fig. 5). Spectral scans of each injected TA muscle using 488 nm laser, 532 nm laser, white light, and background measurements, without laser power correction, were obtained sequentially using custom LabVIEW software. Automated removal of background and autofluorescence from the raw data, and correction for reflectance and laser power, enabled precise determination of GFP and DsRed spectra (Fig. 4d). DsRed/GFP values were determined automatically by dividing the fitted spectral amplitude of corrected DsRed fluorescence using the 532 nm laser by the corrected GFP fluorescence using the 488

nm laser (Fig. 4d). Using the pure DsRed and GFP spectra obtained using the plasmid DNA injections, we measured spectra in gastrocnemius and lumbar paraspinal muscles of TR transgenic, TR;HSA[LR] bi-transgenic, and wild-type mice (Fig. 4e; Supplementary Fig. 5). DsRed/GFP measurements by spectroscopy appeared similar to values obtained by fluorescence microscopy and showed strong correlation (Fig. 4d, e; Supplementary Fig. 5).

**Comparison of in vivo fluorescence measurements**. Systemic delivery of an ASO designed to induce RNase H cleavage of

**Fig. 4** Calibration of in vivo fluorescence spectroscopy measurements. To calibrate our custom built spectroscopy system, we injected DsRed plasmid DNA into the left TA and GFP plasmid DNA into the right TA of FVB WT mice ($N = 2$), and measured fluorescence by microscopy and spectroscopy. **a** Diagram of the spectroscopy system. Three light sources are used: a 488 nm laser for excitation of GFP, a 532 nm laser for excitation of DsRed, and white light for measurement of reflectance. Separate optical fibers are used for detection of fluorescence excitation and emission. **b** Photograph of a mouse under general anesthesia (isoflurane) in the prone position for spectral measurements of the paraspinal or gastrocnemius muscles. **c** Brightfield and fluorescence microscopy images of the TA muscles in an anesthetized mouse 7 days after injection of each plasmid. The mouse is supine. Bars = 4 mm. **d** Representative uncorrected fluorescence spectra using the 488 nm laser (upper left) and the 532 nm laser (upper middle), and white light reflectance (upper right) in TA muscles injected with GFP (black circles) and DsRed (blue triangles) plasmids, or received no plasmid (yellow diamonds). Corrected fluorescence spectra (lower left and lower middle) are calculated by subtraction of background values, division by reflectance values, normalization of laser power, and the use of singular value decomposition (SVD) to separate the GFP and DsRed spectra and remove autofluorescence (Methods). DsRed/GFP ratios (lower right) are calculated automatically by LabView software using peak SVD corrected DsRed emission with the 532 nm laser and GFP emission with the 488 nm laser. ***$P = 0.0002$; one-way ANOVA. **e** We used spectroscopy to measure in vivo fluorescence in gastrocnemius muscles of TR;HSA[LR] bi-transgenic (black circles) ($N = 4$ mice; 8 muscles total), TR transgenic (blue triangles) ($N = 5$ mice; 10 muscles total), and WT control (yellow diamonds) ($N = 1$ mouse; 2 muscles total). Shown are representative uncorrected fluorescence, reflectance, and corrected fluorescence spectra of each, and DsRed/GFP ratios for all muscles examined. For comparison, in vivo fluorescence microscopy data are shown in Supplementary Figure 5. ****$P < 0.0001$; one-way ANOVA

pathogenic transcripts can correct splicing defects and reverse the phenotype in HSA[LR] mice by targeting *ACTA1*-CUG[exp] transcripts[8]. To compare in vivo fluorescence microscopy and spectroscopy measurements of ASO activity, we treated TR;HSA[LR] mice with the ASO from the previous study[8], 445236, by subcutaneous injection and performed weekly fluorescence measurements using each method. Based on DsRed/GFP values, a therapeutic effect was evident in gastrocnemius and lumbar paraspinal muscles as early as Day 14 (four total doses) using each method, and became progressively more pronounced, continuing even after the eighth and final dose on Day 25 (Fig. 5a, b; Supplementary Fig. 6). Although the precise DsRed/GFP values obtained by spectroscopy and microscopy were different, the measurements showed a strong correlation throughout the treatment course (Fig. 5c). RT-PCR analysis of muscles harvested at Day 42 confirmed exon 22 splicing correction of both the transgene *ATP2A1* and endogenous *Atp2a1* (Fig. 5d, e). Automated spectroscopy measurements of DsRed/GFP ratios saved considerable analysis time over manual drawing of ROIs, background subtraction, and calculation of ratios that are required with imaging (Table 1).

**Novel ligand-conjugated antisense oligonucleotide**. Although ASOs are effective in skeletal muscle, drug activity is less robust than in other tissues such as liver, which may be due to a combination of poor tissue bioavailability and insufficient potency[27]. Ligand-conjugated antisense (LICA) chemistry adds specific conjugates to ASOs that are designed to increase drug uptake. In a recent clinical trial, a LICA oligo targeting apolipoprotein(a) transcripts in the liver was several-fold more potent than the unconjugated parent ASO, enabling a >10-fold lower dose and improving tolerability[28]. To determine whether this approach could be adapted for use in muscle tissue, we tested a LICA modified ASO, 992948, that contains a C16 fatty acid conjugate and targets the identical sequence in *ACTA1*-CUG[exp] transcripts in the HSA[LR] mouse model as the unconjugated parent ASO 445236 described in the previous study[8]. Using serial in vivo spectroscopy measurements, the LICA oligo (25 mg/kg twice weekly for 4 weeks; eight total doses = 200 mg/kg total ASO) demonstrated an increase in DsRed/GFP ratios in gastrocnemius and lumbar paraspinal muscles as early as Day 7, after only two doses (Fig. 6a; Supplementary Fig. 7). Activity of the LICA oligo was dose-dependent in both muscles, although was evident earlier in lumbar paraspinal muscles than in gastrocnemius. In the gastrocnemius, peak DsRed/GFP values were obtained by Day 21 (six doses), but continued to increase in lumbar paraspinal

muscles until maximum effect was observed at Day 35, or 10 days after the eighth and final dose. Subcutaneous injection of saline had no effect. Due to an absence of drug activity by Day 28 in gastrocnemius muscles of mice treated with the 2.5 mg/kg and 8.3 mg/kg doses, we continued treatment of these mice by administering an additional 4 doses over the next 2 weeks, for a total of 12 doses (8.3 mg/kg×12 doses = 99.6 mg/kg total; 2.5 mg/kg×12 doses = 30 mg/kg total). By Day 42, DsRed/GFP values in gastrocnemius muscles of mice treated with the 8.3 mg/kg dose were increased as compared to saline-treated mice. Droplet digital PCR (ddPCR) quantitation of *ACTA1*-CUG[exp] transcript levels in muscles collected at Day 42 demonstrated a dose-dependent reduction of the ASO target (Fig. 6b). Inclusion of transgene (Tg) and endogenous mouse *Atp2a1* exon 22 by RT-PCR also showed dose-dependent correction (Fig. 6c, d), similar to the correction of DsRed/GFP values evident by spectroscopy. Splicing of several additional transcripts also showed dose-dependent correction in treated mice (Supplementary Fig. 8).

To compare drug target engagement in muscle tissue of an ASO with and without a LICA-modification, we treated TR;HSA[LR] mice with 12.5 mg/kg twice weekly for 4 weeks of either LICA oligonucleotide 992948 or the unconjugated parent ASO 445236 vs. saline, and monitored fluorescence by serial in vivo spectroscopy. In mice treated with the LICA oligo, an increase of DsRed/GFP measurements was evident in gastrocnemius muscles by Day 14 and in lumbar paraspinal muscles by Day 7, while in mice treated with the unconjugated ASO, the increase of DsRed/GFP values was delayed to Day 28 in gastrocnemius and Day 14 in lumbar paraspinal muscles (Fig. 7a, b). Using ddPCR, knockdown of *ACTA1* transcript levels measured at Day 28 was significantly greater in gastrocnemius, lumbar paraspinal, and quadriceps muscles of mice treated with the LICA oligo than in mice treated with the unconjugated ASO (Fig. 7c). Similarly, RT-PCR analysis demonstrated that exon 22 inclusion of the transgene *ATP2A1* and endogenous mouse *Atp2a1* was significantly higher in gastrocnemius and lumbar paraspinal muscles examined of mice treated with the LICA oligo as compared to the unconjugated ASO (Fig. 7d, e). Differential activity between the LICA and unconjugated ASO was dose-dependent and maintained through Day 42 of treatment (Supplementary Fig. 9). In the lumbar paraspinal muscles, the 12.5 mg/kg dose of the LICA oligo demonstrated pharmacological activity that was equal to or better than the 25 mg/kg dose of the unconjugated ASO throughout the course of treatment, suggesting it is at least two-fold more potent (Supplementary Fig. 9). Fluorescence spectroscopy measurements of DsRed/GFP ratios in

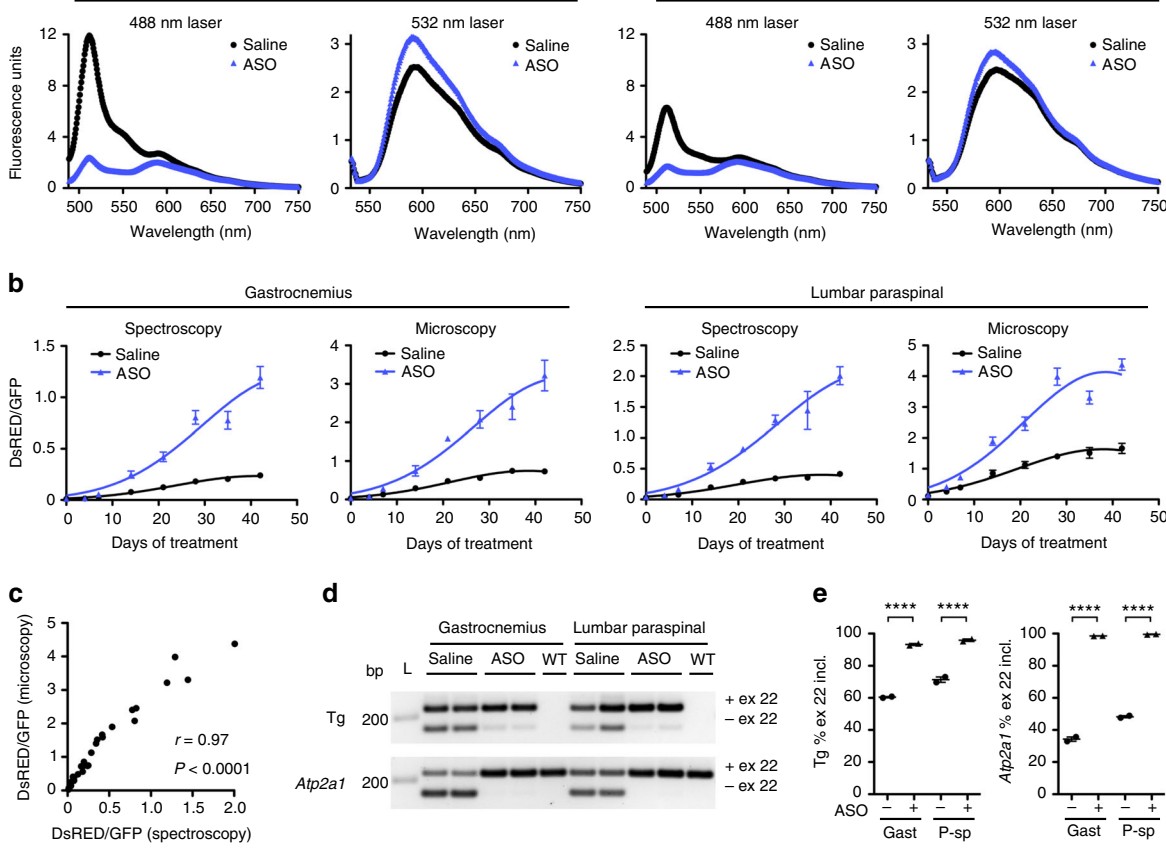

**Fig. 5** Comparison of in vivo fluorescence microscopy and spectroscopy. We treated TR;HSA^LR bi-transgenic mice with saline or ASO 445236 (*N* = 2 each) by subcutaneous injection (25 mg/kg twice weekly for 4 weeks)[8] and monitored DsRed and GFP fluorescence using in vivo spectroscopy and microscopy for 6 weeks. **a** Representative corrected fluorescence spectra, excited at 488 nm and 532 nm in gastrocnemius (left) and lumbar paraspinal (right) muscle 42 days after beginning treatment with saline (black circles) or ASO (blue triangles). **b** Serial in vivo fluorescence spectroscopy and microscopy measurements through 42 days showing ratio of fitted DsRed fluorescence magnitude to fitted GFP fluorescence magnitude in gastrocnemius and lumbar paraspinal muscles of TR;HSA^LR bi-transgenic mice treated with saline (black circles) or ASO (blue triangles). Error bars indicate mean ± s.e.m. Non-linear regression. **c** Correlation of DsRed/GFP values as measured by spectroscopy (*x*-axis) and microscopy (*y*-axis) at each time point. The correlation coefficient *r* and *P* value are shown. **d** Exon 22 splicing analysis by RT-PCR of transgene (Tg) and endogenous mouse *Atp2a1* in gastrocnemius and lumbar paraspinal muscles of saline- and ASO-treated mice at Day 42. L DNA ladder, bp base pairs. **e** Quantitation of the splicing results in **d**. Error bars indicate mean ± s.e.m. ****P < 0.0001, saline vs. ASO; two-way ANOVA

**Table 1 The time required, in minutes (min), to quantitate the DsRed/GFP ratio using spectroscopy or microscopy in left and right gastrocnemius and left and right lumbar paraspinal muscles of *N* = 4 mice (see Fig. 5)**

| Imaging day | Spectroscopy (min) | Microscopy: user 1 (min) | Microscopy: user 2 (min) | Microscopy: mean (min) |
|---|---|---|---|---|
| 28 | 0 | 34 | 27 | 30.5 |
| 35 | 0 | 29 | 26 | 27.5 |
| 42 | 0 | 26 | 21 | 23.5 |

All spectroscopy values are zero because the data acquisition software (LabVIEW) corrects for laser power and acquisition time, subtracts background fluorescence, and calculates each ratio automatically at the end of each scan. After acquisition of microscopy images, quantitation of fluorescence requires manual drawing of regions of interest (ROIs), export of the data from each image into a comma separated values (CSV) file, correction for exposure time, subtraction of background fluorescence, and calculation of DsRed/GFP fluorescence ratio (see Supplementary Fig. 3). Values in minutes for two separate users, and the mean of the two, are shown for the images obtained on Days 28, 35, and 42.

gastrocnemius and lumbar paraspinal muscles showed good correlation with RT-PCR analysis exon 22 inclusion of both the transgene *ATP2A1* and endogenous mouse *Atp2a1* in these muscles (Fig. 8a, b).

ASO drug concentrations in muscle and liver correlated with total dose administered, were significantly higher in paraspinal muscles than in gastrocnemius muscles, and were higher in liver

than in either gastrocnemius or paraspinal muscles (Supplementary Fig. 9). To determine whether ASO drug activity and concentration in muscles may be related to vascular supply, we examined capillary density and found that paraspinal muscles have higher number of capillaries per muscle fiber than gastrocnemius muscles (Fig. 9a, b). Unexpectedly, we also found that capillary density was greater in saline-treated TR;HSA^LR

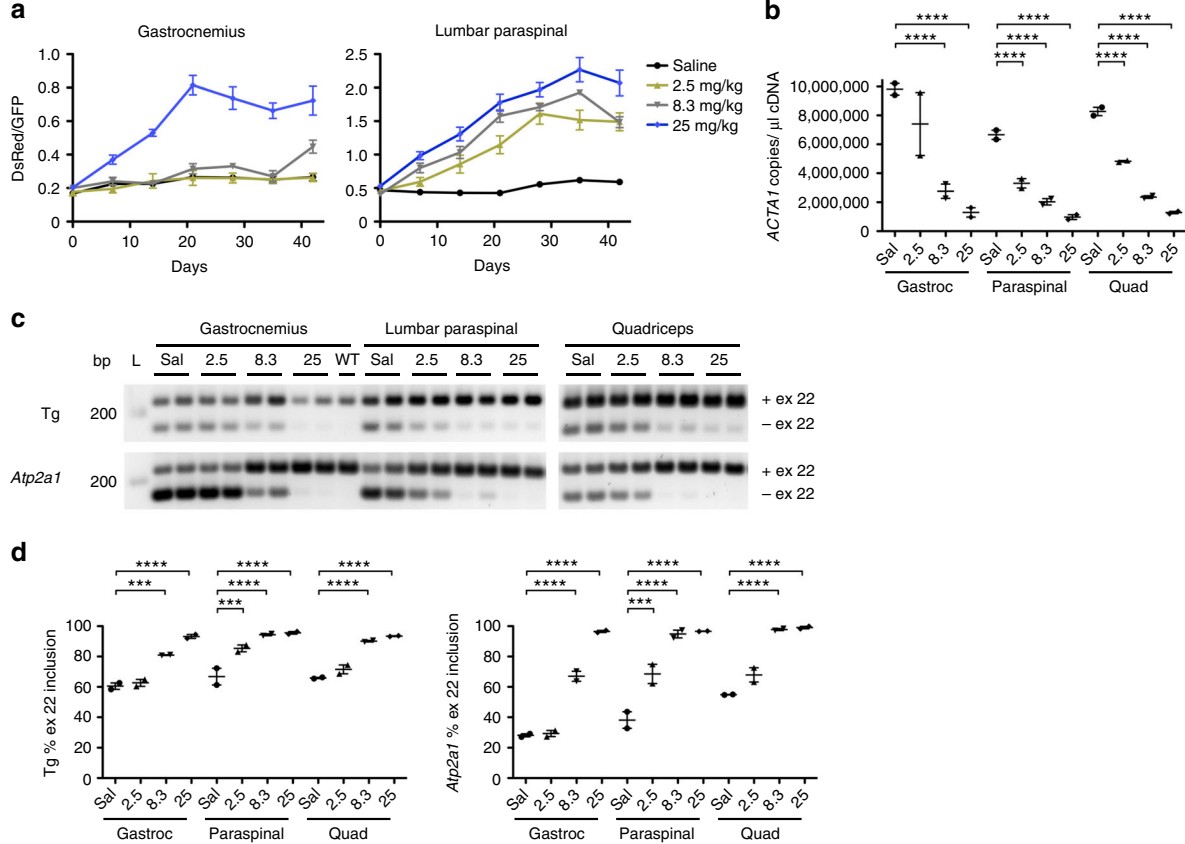

**Fig. 6** In vivo activity of a novel ligand-conjugated antisense (LICA) oligonucleotide. We treated TR;HSA^LR bi-transgenic mice with saline or a novel LICA oligonucleotide, ASO 992948, targeting *ACTA1*-CUG^exp transcripts and measured DsRed and GFP fluorescence by serial in vivo spectroscopy. LICA-oligo doses were 2.5, 8.3, or 25 mg/kg twice weekly for 4 weeks (8 total doses) by subcutaneous injection ($N = 2$ each group). However, due to low DsRed/GFP fluorescence measurements in gastrocnemius muscles, mice receiving the 2.5 mg/kg and 8.3 mg/kg doses received four additional injections over the next 2 weeks, for a total of 12 doses. **a** Serial DsRed/GFP measurements in gastrocnemius and lumbar paraspinal muscles through Day 42. The final dose in the saline (black circles) and 25 mg/kg groups (blue diamonds) was Day 24, while the final dose in the 2.5 (yellow triangles) and 8.3 mg/kg (gray triangles) groups was Day 37. Error bars indicate mean ± s.e.m. **b** Quantitation of *ACTA1*-CUG^exp transcript levels (copies per microliter cDNA) by droplet digital PCR in muscles collected at Day 42. Error bars indicate mean ± s.e.m. ****$P < 0.0001$; two-way ANOVA. **c** Splicing analysis by RT-PCR in gastrocnemius and lumbar paraspinal muscles collected at Day 42. L DNA ladder, bp base pairs. **d** Quantitation of splicing results in **c**. Error bars indicate mean ± s.e.m. ****$P < 0.0001$, ***$P < 0.001$; two-way ANOVA

mice than in WT controls, and that ASO treatment for 42 days was associated with a reduction of capillary density toward WT values (Fig. 9a, b). Histologic analysis of ASO-treated muscles showed a decrease in the percentage of muscle fibers containing internal nuclei, an improvement of muscle fiber diameter measurements toward the WT pattern, and no evidence of toxicity (Fig. 9c, d).

## Discussion

We describe a novel TR;HSA^LR bi-transgenic mouse model that expresses an alternative splicing reporter of DM1 disease activity and enables a convenient non-invasive estimation of in vivo pharmacodynamic properties. This model will be useful for rapid identification of candidate therapeutics for reducing pathogenicity of CUG^exp transcripts in DM1, including new ASO chemistries and conjugates, small molecules, short interfering RNAs (siRNAs), gene therapy vectors for the production of antisense RNAs, protein-based therapies that rescue aberrant splicing, and gene editing approaches that reduce the genomic CTG repeat length or inhibit transcription of CUG^exp repeats[29–39]. This model also is well suited to meet an equally important goal of drug development tools: rejection of failed drugs and therapeutic

strategies at an early stage, thereby saving valuable resources before proceeding to costly clinical trials. We propose that the TR;HSA^LR bi-transgenic model replace the HSA^LR single transgenic model for in vivo screening of new candidate therapeutics because it has all the advantages of that model, while also reducing the time needed to identify candidate therapeutics with the most promise to proceed to clinical trials, and meeting an Institutional Animal Care and Use Committee (IACUC) ethical goal of limiting the number of mice needed.

This binary splicing switch determines the expression of either DsRed or GFP, and provides a highly sensitive and quantitative measure of splicing outcomes by the ratio of these fluorescent proteins, independent of overall reporter gene expression level. Regarding precision, the fluorescence readout is a ratio of two signals (DsRed vs. GFP) both generated from the same construct. This design eliminates an important source of biological variation (differences in level of reporter gene expression) and reduces measurement error (a ratio of two fluorescence signals is less dependent on background and detection than absolute levels of a single signal). Regarding validity, the fluorescence readout directly reports developmental alternative splicing, the normal outcome in muscle repair[16] and the fundamental biochemical derangement in DM1, which is

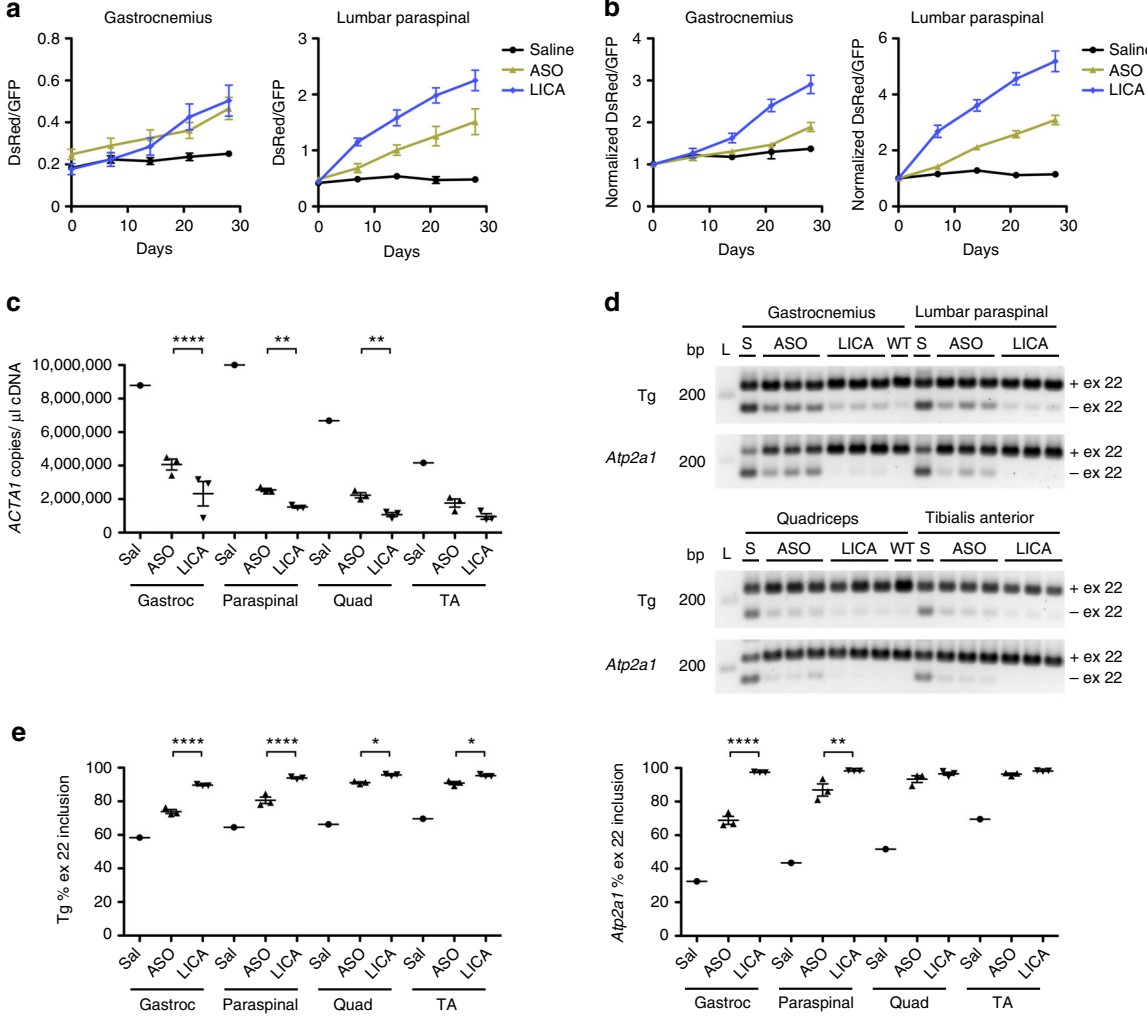

**Fig. 7** In vivo comparison of LICA and the unconjugated parent ASO. We treated TR;HSA[LR] mice with LICA oligo 992948 (LICA) or ASO 445236 (ASO), which is the unconjugated parent of LICA oligo 992948 that targets the identical *ACTA1* sequence[8]. ASOs were administered by subcutaneous injection of 12.5 mg/kg twice weekly for 4 weeks (*N* = 3 each) (eight total doses). Treatment with saline (*N* = 1) served as a control. **a** DsRed/GFP quantitative fluorescence in gastrocnemius (left) and lumbar paraspinal muscles (right) by serial in vivo spectroscopy in mice treated with saline (black circles), ASO (yellow triangles), or LICA (blue diamonds). Error bars indicate mean ± s.e.m. **b** Due to the higher baseline DsRed/GFP values in the gastrocnemius of mice randomized to receive the unconjugated ASO, we normalized each measurement of quantitative fluorescence in the gastrocnemius (left) and lumbar paraspinal muscles (right) to the Day 0 value, so that the Day 0 value for each muscle = 1. Error bars indicate mean ± s.e.m. **c** ddPCR quantitation of *ACTA1* transcripts (copies per microliter cDNA) in muscles collected at imaging Day 28. ****$P < 0.0001$; **$P < 0.01$; two-way ANOVA. **d** RT-PCR analysis of exon 22 alternative splicing of the transgene (Tg) and mouse *Atp2a1* in gastrocnemius, lumbar paraspinal, quadriceps, and tibialis anterior (TA) muscles collected on imaging Day 28. FVB WT gastrocnemius served as a control. L DNA ladder, bp base pairs. **e** Quantitation of splicing results in **d**. Error bars indicate mean ± s.e.m. ****$P < 0.0001$; **$P < 0.01$; *$P < 0.05$; two-way ANOVA

clearly linked to production of symptoms[5,6,40,41]. Regarding throughput, the fluorescence measurements enable a short and simple assay. The restriction of DsRed protein expression to the cytoplasm and GFP expression to myonuclei suggests the possibility that the *N*-terminal peptide sequence upstream of GFP may act as a nuclear localizing signal.

In this study, we also demonstrate that the TR-*ATP2A1* construct enables non-invasive tracking of muscle regeneration in live mice, suggesting it also can serve as a more general biomarker of muscle regeneration, and potentially for muscle repair in other muscular dystrophies beyond DM1. The delayed upregulation of the DsRed/GFP ratio following muscle injury in TR mice may be related to the transition from immature DsRed monomers that fluoresce mostly green to mature tetramers that fluoresce red, a process that takes several days[42]. The similarity of splicing changes in DM1 and DM2[5] suggests that the TR model could be

used as a drug development tool for DM2 as soon as a robust mouse model of DM2 becomes available.

Results obtained using in vivo fluorescence microscopy and spectroscopy showed strong correlation, suggesting that either method can be used for non-invasive detection of therapeutic response. A major advantage of our fluorescence spectroscopy system over conventional fluorescence microscopy is the speed of analysis: the quantitative DsRed/GFP ratios, with all corrections for background, autofluorescence, and SVD fitting, are calculated automatically at the end of each spectroscopy scan. By contrast, quantitation of fluorescence in microscopy images is calculated by hand, which involves the drawing of individual ROIs, subtraction of background from each channel, correction for image exposure time, and calculation of DsRed/GFP ratios, which typically takes several minutes per mouse. As a result, throughput is significantly higher with spectroscopy.

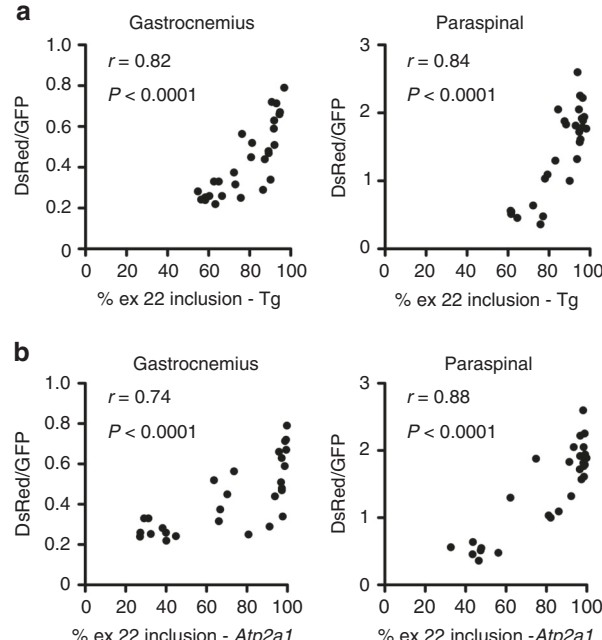

**Fig. 8** Correlation of DsRed/GFP measurements with splicing in muscle tissue. We used fluorescence spectroscopy to determine the DsRed/GFP in gastrocnemius and lumbar paraspinal muscles of TR;HSA$^{LR}$ bi-transgenic mice treated by subcutaneous injection of saline, unconjugated ASO, or LICA oligo (N = 28) (see Figs. 6 and 7, and Supplementary Figs. 7–9). After the final spectroscopy measurements at either Day 28 or 42, we dissected the gastrocnemius and lumbar paraspinal muscle tissues. To validate the spectroscopy measurements as estimations of alternative splicing in the entire muscle, we correlated final DsRed/GFP measurements in gastrocnemius and lumbar paraspinal muscles with RT-PCR determination of exon 22 inclusion of **a** the *ATP2A1* transgene (Tg) and **b** endogenous mouse *Atp2a1* in these muscles. The correlation coefficient r and P value for each are shown

Advantages of this in vivo spectroscopy system over similar earlier instruments[24–26] include greater sensitivity due to dedicated GFP and DsRed lasers, and faster data acquisition and analysis due to automated control of filter and fiber switching. Fluorescence spectroscopy also offers a more sensitive ratiometric analysis than imaging approaches, due in part to more accurate separation of overlapping emission spectra (such as GFP and DsRed) in the presence of an autofluorescence background. An additional advantage of spectroscopy includes improved signal-to-noise ratio for detection of low DsRed and GFP levels at very early and late stages of the treatment. Consequently, we expect this spectroscopy system may enable more accurate tracking of disease progression or regression by detection of relatively low levels of fluorophores that may be beyond detection by fluorescence imaging. The greater sensitivity of spectroscopy also may broaden the potential use of TR constructs as biomarkers for other applications. For example, by modifying the minigene, the TR construct may be useful as a drug development tool for other disorders that are candidates for RNA modulation therapies beyond DM1, including Duchenne muscular dystrophy, Hutchinson-Gilford progeria syndrome, and amyotrophic lateral sclerosis[43–45].

In this study, we also show that a gapmer C16 fatty acid LICA oligonucleotide achieves efficient dose-dependent target knockdown in skeletal muscle and that concentrations of the LICA oligo in muscle tissue were approximately two-fold higher than the unconjugated parent ASO after a 4-week course of treatment. The more rapid increases in DsRed/GFP values and the more robust splicing correction in mice treated with the LICA oligo suggests that the C16 ligand enables greater target engagement in muscle tissue than the unconjugated ASO. Non-invasive measurements also revealed a more rapid ASO target engagement in paraspinal muscles as compared to gastrocnemius muscles, which may be related to the higher ASO concentrations and the greater capillary density that we observed in paraspinal vs. gastrocnemius muscles. The greater capillary density in TR;HSA$^{LR}$ than WT mice may be related to dysregulation of genes involved in angiogenesis and blood vessel maturation[46], while the reduction of capillary density in ASO-treated mice may enable a novel measure of therapeutic response in muscle tissue. Our data support further development of LICA technology for treatment of DM1 and other for ASO applications targeting skeletal muscle.

## Methods

**Alternative splicing therapy reporter (TR) construct**. We modified the bi-chromatic RG6 construct[11] (a gift of Dr. T. Cooper) by replacing the chicken *TNNT2* exon 5 minigene with a human *ATP2A1* exon 22 minigene into the *Xba* I/*Age* I site to create the fluorescence bi-chromatic FBR-*ATP2A1* reporter construct. To bypass a stop codon and enable shift of the reading frame[13], exon 22 of the *ATP2A1* minigene was modified by site-directed mutagenesis to induce a single base pair deletion so that it contains 41 base pairs instead of 42 (a gift of Dr. C. Thornton). In designing the construct for in vivo use, we were concerned with several previous reports of dose-dependent in vivo toxicity of GFP expression in skeletal and cardiac muscle, whether by transgenic expression or viral vector-mediated delivery[47–53], as well as unpublished observations (T.M.W.). To reduce the possibility that therapeutic ASOs, or other treatments inhibiting CUG$^{exp}$ RNA pathogenicity, could exacerbate the myopathy by upregulating GFP expression, the reading frame was designed so that, in contrast to the chicken *TNNT2* exon 5 minigene in the RG6 construct, inclusion of the *ATP2A1* minigene exon 22, which is high in normal mature muscle, would result in the DsRed reading frame instead of GFP (Fig. 1a).

Next, we removed the CMV promoter, FLAG and NLS regions using a *Bgl* II/*Xba* I double digest and replaced it with a custom linker containing *Cla* I, *Not* I, Kozak, ATG, and FLAG sequences. To restrict expression of the construct to skeletal muscle, we removed the human skeletal actin (HSA) promoter/enhancer/vp1 splice acceptor region[14] from the pBSx-HSAvpA plasmid (a gift of Dr. J. Chamberlain via Dr. A. Burghes) and cloned it into the *Cla* I/*Not* I site to create the new HSA-FLAG-ATP2A1ex22-DsRed-GFP TR reporter construct. The sequence of the TR construct is shown in Supplementary Note 1.

**Experimental mice**. Institutional Animal Care and Use Committees (IACUCs) at Massachusetts General Hospital and University of Rochester approved all studies in mice described here. The Human Skeletal Actin - Long Repeat (HSA$^{LR}$) mouse model of DM1 expresses a human *ACTA1* transgene that contains ~220 CTG repeats in the 3′ untranslated region[3]. TR reporter transgenic mice were bred with HSA$^{LR}$ transgenic mice, both on the FVB background, to create TR;HSA$^{LR}$ bi-transgenic mice. FVB wild-type mice served as controls.

**Intramuscular injection of plasmid DNA**. At each stage of cloning, we tested constructs for in vivo expression by intramuscular injection and electroporation[6] of plasmid DNA (20 or 25 μg) in the tibialis anterior (TA) muscle of HSA$^{LR}$ and wild-type mice, and determined gene expression by non-invasive in vivo fluorescence microscopy, splicing of TR-*ATP2A1* and endogenous *Atp2a1* mouse transcripts by RT-PCR, and examination of muscle cryosections by fluorescence microscopy.

**Generation of novel TR transgenic mice**. We digested the TR plasmid with *Bgl* II/*Sph* I restriction enzymes, gel extracted, and purified the 5.316 kb linear DNA fragment containing the HSA-TR construct. A commercial vendor (Cyagen Biosciences) microinjected HSA-TR linear DNA and generated three transgenic founders on an FVB background, two of which produced offspring. One of these lines expressed the transgene at a level sufficient to visualize DsRED and GFP fluorescence by multiple non-invasive means.

**Genotyping offspring**. We identified mice by toe clipping[54,55] on postnatal day 7 and used each toe biopsy to isolate DNA using a tissue/blood DNA isolation kit (Qiagen). To detect the TR transgene, we used PCR analysis of DNA with two sets of genotyping primers, one specific for DsRed, the other for GFP, as follows:
  DsRED left primer: 5′-GGCCACAACACCGTGAAGC-3′
  DsRED right primer: 5′-CGCCGTCCTCGAAGTTCATC-3′
  GFP left primer: 5′-TGCAGTGCTTCAGCCGCTAC-3′
  GFP right primer: 5′-CTGCCGTCCTCGATGTTGTG-3′

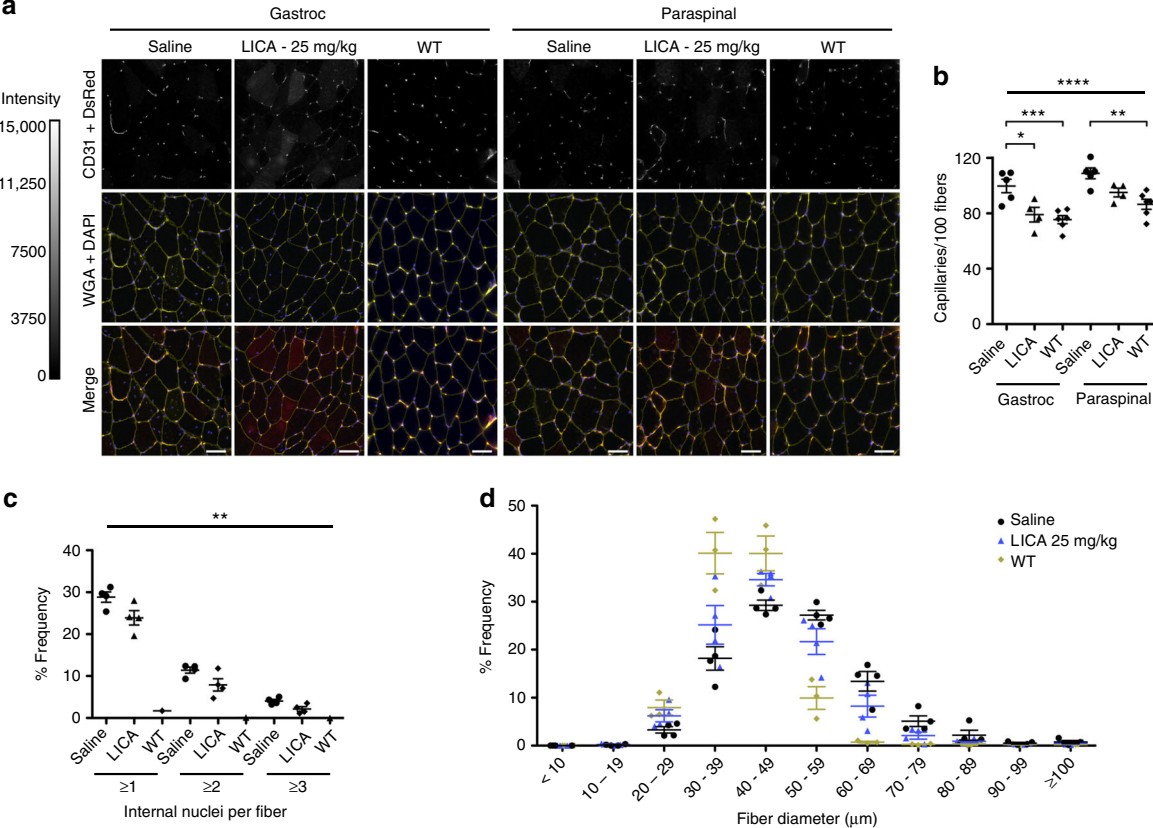

**Fig. 9** LICA oligonucleotide treatment effects. **a** We examined capillary density in TR;HSA[LR] mice treated with saline ($N = 5$) or LICA oligonucleotide 992948 (25 mg/kg twice weekly for 4 weeks; $N = 4$) by immunolabeling using an anti-CD31 antibody. Untreated FVB wild-type (WT) mice ($N = 6$) served as controls. Shown are representative images of gastrocnemius muscle cryosections. Muscle fibers are highlighted with Alexa 647-wheat germ agglutinin (WGA; pseudocolored yellow) and nuclei with DAPI (blue). Merge DsRed (red) + GFP (green) + WGA + DAPI. Fluorescence images are the extended focus of deconvolved Z-series. Intensity scale = 0–15,000 grayscale units (CD31). Bars = 50 μm. **b** Quantitation of capillary density, as measured by the number of capillaries per 100 gastrocnemius or paraspinal muscle fibers, in TR;HSA[LR] mice treated with saline or LICA oligo, and untreated WT controls. Error bars indicate mean ± s.e.m. ****$P < 0.0001$, overall difference between saline-treated, LICA-treated, and WT groups; ***$P < 0.001$ (gastrocnemius, saline-treated vs. WT); **$P < 0.01$ (paraspinal, saline-treated vs. WT), and $P = 0.0012$, (overall difference between gastrocnemius and paraspinal); *$P < 0.05$ (gastrocnemius, saline-treated vs. LICA-treated); two-way ANOVA. **c** Quantitation of internal nuclei frequency in the saline-treated, LICA-treated, and WT groups. Error bars indicate mean ± s.e.m. **$P = 0.0015$ saline vs. treated group; two-way ANOVA. **d** Minimum Feret's diameter, defined as the minimum distance between parallel tangents[62], of gastrocnemius muscle fibers of TR mice treated with saline (black circles) or the LICA oligo (blue triangles), and untreated wild-type (WT) controls (yellow diamonds). Error bars indicate mean ± s.e.m

Breeding of TR hemizygotes (TR[+/−]) with WT mice (TR[−/−]) produced TR[+/−] offspring at the expected 50% frequency. The absence of detectable in vivo fluorescence in TR[+/−] hemizygotes required generation of TR homozygotes (TR[+/+]), which were obtained at the expected 25% frequency by breeding TR[+/−] hemizygotes. To generate TR;HSA[LR] bi-transgenic mice, we crossed TR[+/−] mice with homozygous HSA[LR] mice (HSA[LR+/+]), generating TR[+/−];HSA[LR+/−] double hemizygotes, then crossed TR[+/−] mice with TR[+/−];HSA[LR+/−] double hemizygotes, generating TR[+/+];HSA[LR+/−], and finally crossed TR[+/+];HSA[LR+/−] with TR[+/+];HSA[LR+/−] to generate TR[+/+];HSA[LR+/+] double homozygous bi-transgenic mice that were used for all imaging experiments. We determined zygosity of the TR transgene using the ratio of DsRed/Acta1 transcripts, and zygosity of the ACTA1 transgene using the ratio of ACTA1/Acta1 transcripts, as measured by RT-PCR and quantitative band densitometry (Supplementary Fig. 3; Supplementary Table 1).

**In vivo fluorescence imaging**. We removed hair (Nair) and performed all in vivo imaging under general anesthesia consisting of either inhalation isoflurane 1–3% to effect or a cocktail of ketamine 100 mg/kg, xylazine 10 mg/kg, and acepromazine 3 mg/kg by intraperitoneal injection[6]. We imaged mice using an AxioZoom fluorescence microscope (Zeiss), ×0.5 and ×1.0 objectives, separate filters for GFP (excitation/emission 470/525; Zeiss filter set 38 HE) and DsRed (excitation/emission 550/605; Zeiss filter set 43 HE), an ORCA R2 CCD camera (Hamamatsu), and Volocity image acquisition software (Perkin Elmer). To quantitate fluorescence, we measured DsRed and GFP fluorescence in regions of interest (ROIs) using the Volocity quantitation software module, subtracted background fluorescence,

corrected for exposure time, and calculated DsRed/GFP fluorescence ratios. Alternatively, mice were imaged using the IVIS Spectrum (Perkin Elmer) with automatic exposure sequences for excitation/emission 465/520 nm (GFP) and 535/600 nm (DsRed), and quantitated fluorescence in ROIs using Living Image software (Perkin Elmer).

**Acute muscle injury**. Using a standard muscle injury paradigm[17], we injected 30 μl of 1.2% barium chloride (BaCl₂) into gastrocnemius muscles of TR transgenic mice and determined DsRED/GFP quantitative fluorescence by serial in vivo imaging. FVB wild-type mice served as non-fluorescent controls. Contralateral gastrocnemius or TA muscles either were untreated or injected with saline.

**Antisense oligonucleotides**. We purchased 25-mer CAG repeat morpholino[7] (CAG25; 5′-AGCAGCAGCAGCAGCAGCAGCAGCA-3′) antisense oligonucleotides (ASO) containing phosphorodiamidate internucleotide linkages and an octaguanidine dendrimer conjugated to the 3′ end of the oligo[56] (Gene Tools, LLC). ASO 445236 and ASO 992948 are 20-mer RNase H-active gapmer ASOs, wherein the central gap segment consists of ten 2′-deoxyribonucleotides that are flanked on the 5′ and 3′ wings by five 2′-O-methoxyethyl-modified nucleotides. The sequence of ASO 992948 is identical to ASO 445236, but includes the addition of a C16 lipophilic group conjugated to the 5′ terminus via a hexylamino (HA) linker that has a phosphodiester (P₀) at the 3′ end. The internucleotide linkages are phosphorothioate, and all cytosine residues are 5′-methylcytosines.

**Gapmer ASO sequences**. 445236: 5′-CCATTTTCTTCCACAGGGCT-3′ (published previously[8])

992948: 5′-C16-HA-P₀-CCATTTTCTTCCACAGGGCT-3′

We administered ASOs locally into the gastrocnemius and/or lumbar paraspinal muscles of TR;HSA^LR bi-transgenic mice by intramuscular injection under general anesthesia without electroporation, or systemically by subcutaneous injection under light restraint[8], and followed treatment effects by serial in vivo fluorescence microscopy and/or spectroscopy under general anesthesia.

**Quantification of ASO tissue concentration**. ASO was extracted from muscle tissue by phenol-chloroform followed by a solid-phase extraction, and quantified by high performance liquid chromatography coupled with tandem mass spectrometry detection[57].

**RNA isolation**. We homogenized muscles in Trizol (Life Technologies), removed DNA and protein using bromochloropropane, precipitated RNA with isopropanol, washed pellets in 75% ethanol, and dissolved pellets in molecular grade water according to manufacturer recommendations. To determine RNA concentration and quality, we measured A260 and A280 values (Nanodrop) and examined 18 S and 28 S ribosomal RNA bands by agarose gel electrophoresis.

**RT-PCR analysis of alternative splicing**. We made cDNA using Superscript II reverse transcriptase (Life Technologies) and oligo dT, and performed PCR using Amplitaq Gold (Life Technologies) and gene-specific primers. We separated PCR products using agarose gels, labeled DNA with 1x SYBR I green nucleic acid stain (Life Technologies), and quantitated band intensities using a transilluminator, CCD camera, XcitaBlue™ conversion screen, and Image Lab image acquisition and analysis software (Bio-Rad). We designed primers for TR-*ATP2A1*, *Mbnl2*, and *Vps39* using Primer3 software[58,59]. Primers for *Clcn1*, *m-Titin*, *Clasp1*, *Map3k4*, *Mbnl1*, *Ncor2*, *Nfix*, and endogenous mouse *Atp2a1* were published previously[2,6,60]. A complete list of primer sequences is shown in Supplementary Table 4. Uncropped gels are shown in Supplementary Figs. 10–13.

**Microscopy of muscle tissue sections**. Muscles were frozen in isopentane cooled in liquid nitrogen. We stained 8 μm cryosections with hematoxylin and eosin. To visualize native DsRED and GFP proteins in muscle tissue, we fixed 8 μm cryosections in 3% paraformaldehyde, washed in 1× PBS, counterstained nuclei with DAPI, and mounted with anti-fade medium (Prolong Gold, Invitrogen product # P36930). To highlight muscle fibers, we counterstained with Alexa 647 wheat germ agglutinin (WGA; 10 μg/ml; Invitrogen product # W32466) or used an anti-laminin antibody (0.5 μg/ml; Abcam product # 14055) and a goat anti-chicken Alexa 647 secondary antibody (1 μg/ml; Invitrogen product # A21449). To identify capillaries, we used an anti-CD31 rabbit monoclonal primary antibody (5.5 μg/ml; Abcam product # EPR17260–263), a goat anti-rabbit Alexa 546 secondary antibody (1 μg/ml; Invitrogen product # A11018), counterstained nuclei with DAPI and Alexa 647 WGA (20 μg/ml; Invitrogen product # W32466), and mounted with anti-fade medium (Prolong Gold). To capture single images or z-series stacks, we used an AxioImager microscope (Zeiss), ×10, ×20, ×40, or ×63 objectives, filters for DAPI (excitation/emission 365/445; Zeiss filter set 49), GFP (excitation/emission 470/525; Zeiss filter set 38), Cy3 (excitation/emission 550/605; Zeiss filter set 43 HE), and Cy5 (excitation/emission 640/690; Zeiss filter set 50), a Flash 4.0 LT sCMOS camera (Hamamatsu), a MicroPublisher 3.3 RTV color CCD camera (Q-Imaging), and Volocity image acquisition software. To quantitate fluorescence, we used Volocity quantitation and restoration software modules (Perkin Elmer).

**Immunoblotting**. We isolated protein from 15–20 cryosections of muscle, 30 μm thick, added RIPA detergent buffer supplemented with 1x protease inhibitor cocktail (Sigma), vortexed, centrifuged, removed solubilized protein to a fresh microfuge tube, and determined protein concentration using the Bradford assay. To separate proteins, we used pre-cast any kD polyacrylamide gels (Biorad), transferred to 0.45 μm nitrocellulose membranes, and blocked with 5% milk. Primary antibodies were anti-GFP rabbit monoclonal (1:1000; Cell Signaling product # 2956) and anti-GAPDH mouse monoclonal (0.1 μg/ml; AbD Serotec product # MCA4739). Secondary antibodies were goat anti-rabbit IRDye 800CW (0.05 μg/ml; Licor product # 926–32211) and goat anti-mouse IRDye 680RD (0.05 μg/ml; Licor product # 926–68070). To determine protein expression, we quantitated band intensities using a laser scanner (Licor Odyssey Cxl) and ImageStudio software (Licor). Uncropped blots are shown in Supplementary Fig. 10.

**In vivo fluorescence spectroscopy**. In order to measure the GFP and DsRed fluorescence, we constructed a dedicated spectroscopy system. The system consists of sources for fluorescence excitation, a custom optical probe, and a portable spectrometer with large dynamic range and low noise for detection of emission spectra. For excitation, we used diode lasers at 488 nm to excite GFP (OBIS 488 nm LX 50 mW, Coherent Inc., Santa Clara, CA), and at 532 nm to excite DsRed (LMX-532S-50-COL-PP, Oxxius S.A., Lannion, France). The output from each laser is filtered by appropriate laser line filters (LL01–488–12.5 and LL01–532–12.5, Semrock Inc., Rochester, NY) and focused into a fiber switch (F-163–10IR,

Piezosystem Jena, Inc., Hopedale, MA) via a fiber coupler (Oz Optics, Ottawa, Ontario). The fiber switch allows for the excitation wavelength to be toggled in order to match the desired fluorophore. The output of the switch is connected to an optical fiber for excitation (M28L01, Thorlabs, Inc., Newton, NJ), with a separate, identical fiber used for collection of emission. The excitation fiber is secured vertically in a custom-machined probe, while the emission fiber is secured to this mount at a 45° angle at a distance of 9.6 mm from the excitation fiber. The emission fiber is connected to a free space laser coupler (Oz Optics) that directs the collected light through a motor-driven filter wheel (FW102C, Thorlabs Inc., Newton, NJ) equipped with long-pass filters tuned to GFP (LP02–488RU-25, Semrock) and DsRed (BLP01–532R-25, Semrock). After filtering, the collected light is focused again into a fiber with a fiber coupler (Oz Optics) and detected by a compact, TE-cooled spectrometer (QE65PRO-FL, Ocean Optics). In addition to the excitation lasers, a broadband white light source (HL-2000, Ocean Optics) is connected to the fiber switch. This allows for performance of white light spectroscopy, which is used to correct detected fluorescence spectra for the effects of varying tissue optical properties[23]. This correction is vital, as it ensures that changes in detected fluorescence are due only to changes in fluorophores, rather than the effects of changes in the intervening tissue. A complete list of components is shown in Supplementary Table 3.

The entire spectroscopy system is enclosed and controlled via a LabVIEW software interface (National Instruments, Austin, TX). At each measurement point, the excitation source is switched through the two laser sources and the white light source, while the filter wheel is simultaneously changed to the appropriate position. After detection, each spectrum is corrected for dark background and wavelength-dependent system response. Fluorescence spectra then are divided by the white light spectrum in order to correct for the effects of optical properties[24,61]. The corrected fluorescence spectra then are fit using singular value decomposition (SVD) in order to separate the individual fluorophores, as well as remove the effects of any native fluorophores that are present in the tissue, similar to prior use for separation of multiple fluorophores from detected fluorescence spectra[24–26]. Prior to scanning, the fiber optic excitation and emission probe is placed ~1.5 cm above the skin overlying the muscle to be analyzed (Fig. 4b).

**Sample size**. The response to ASO treatment in the TR;HSA^LR mice as measured by in vivo imaging or spectroscopy was unknown. Therefore, we were unable to choose a sample size ahead of time to ensure adequate power to measure pharmacodynamic activity. Instead we estimated sample sizes based on RT-PCR splicing patterns of TR-ATP2A1 exon 22 and in vivo spectroscopy measurements of DsRed/GFP values in gastrocnemius muscles of TR single transgenic and TR;HSA^LR bi-transgenic mice. Using RT-PCR, the difference between means of these two groups was 31 and the standard deviations were 1.6 and 1.7, respectively, meaning that a sample size of 1 from each group would provide 98% power to detect a difference with $P < 0.05$. Using spectroscopy measurements, the difference between means of these two groups was 4.2 and the standard deviations 0.02 and 1.0, respectively, indicating that a sample size of 2 from each group ($N = 4$ total) would provide 98% power to detect a difference with $P < 0.05$. Mice ranged from 4 weeks to 4 months of age and were chosen randomly by genotype and stratified for sex to allow an approximately equal number of females and males. Although one or two examiners were blinded to treatment assignments for the subcutaneous ASO studies, the imaging and/or spectroscopy data obtained during the several week course of each experiment were so robust that it essentially identified the saline or low-dose treatment groups from the higher dose treatment groups prior to termination of the experiments.

**Statistical analysis**. For two-group and multi-group comparisons, we used unpaired two-tailed $t$-test or analysis of variance (ANOVA), respectively (Prism software, GraphPad, Inc.). To determine associations of DsRed/GFP values measured by microscopy with those measured by spectroscopy, we used Pearson correlation coefficients. A $P$ value $<0.05$ was considered significant.

## Data availability

All relevant data are available from the authors.

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

## Acknowledgements

We thank the MGH Martinos Center for Biomedical Imaging for providing access to the IVIS Spectrum, and Drs. T. Cooper, C. Thornton, and A. Burghes for providing DNA constructs. NIH R01NS088202 (T.M.W.) supported this work.

## Author contributions

N.H., L.A., and T.M.W. performed experiments and analyzed data. S.M., T.H.F., and T.M.W. designed the study. T.M.B. designed and custom-built the in vivo fluorescence spectroscopy system. T.M.W. wrote the paper. N.H., L.A., T.M.B., S.M., C.F.B., F.R., T.H.F., and T.M.W. analyzed the data, discussed the results, and commented on the manuscript.

## Additional information

**Competing interests:** C.F.B. and F.R. are employees of Ionis Pharmaceuticals. The remaining authors declare no competing interests.

