## [Peer Review File · Nature Communications]

Reviewer #1 (Remarks to the Author):

This manuscript submitted to Nature Communications reports the development of “therapy reporter” (TR) mice model and fluorescence spectroscopy as drug development tools for evaluation of therapeutic oligonucleotide targeting of skeletal muscle. The technology developed is interesting. But it seems the technique reported is limited for evaluation of drugs only for targeting of muscle tissue. Furthermore, the absorption and emission wavelengths of DsRed and GFP are very different and also not in the NIR-window, and both reporters suffer limited tissue penetration capability. These factors make it is very challenge to do the quantification analysis of the imaging signal. The reliability of using DsRed/GFP ratio needs more validation.

Reviewer #2 (Remarks to the Author):

In this paper Hu and colleagues have generated a novel reporter bi-transgenic mouse model of DM1 based on the well-characterized and muscle-restricted HSALR mouse model. They customized a bi-chromatic reporter system in which the expression of either DsRed or GFP is determined by splicing of an upstream minigene containing ATP2A1 exon 22 that is misregulated in DM1. Moreover, they validated a non-invasive in vivo fluorescence spectroscopy system to monitor splicing changes and evaluate therapeutic activity of ASOs, including a new LICA-ASO.

The manuscript is clear and well written. Experimental evidences support the use of the bi-transgenic mouse model (TR:HSALR) in association with in vivo fluorescence spectroscopy as drug development tools for DM1. These combined non-invasive tools will be useful to evaluate muscle-specific efficacy of therapeutic approaches for DM1 using a primary screening based on ATP2A1 exon 22 missplicing correction. However, this reporter model cannot assess effects of potential therapies on other molecular alterations that were also described in DM1 muscles as abnormal activation of CELF1 or GSK3beta. Nevertheless, the proposed tools provide a novel resource for translational research in DM1.

Comments and concerns:

- Because ATP2A1 exon 22 contains stop codons when expressed in frame, authors should clearly explain that the ATP2A1 minigene has been modified to enable expression of DsRed when ATP2A1 exon 22 is included.
- Plasmid experiments (Fig S1c-g) showed that DsRed is decreased in a DM1 context. However it is not clear whether a concomitant increase of GFP is measured!

- An unexpected exclusive nuclear localization of GFP was observed whereas DsRed is cytoplasmic. This point should be addressed.
- Expression of GFP in a DM1 context was confirmed by Western blot in Fig 2e. What about the quantification of DsRed expression in WT and DM1 condition?
- Data in Fig 6 and 7 were generated from 2 mice each group only. How appropriate are the proposed statistical analyses? Authors should include either additional mice or another independent experiment to strength their results and conclusion.
- Typo error in Fig S6a-b: doses in panel "a" (2.5, 8.5 and 25mg/kg) are different from those in panel "b" (8.5, 12.5 and 25mg/kg)!

Reviewer #3 (Remarks to the Author):

This manuscript describes the development and characterisation of a new in vivo model to test potential therapies for myotonic dystrophy. Myotonic dystrophy is caused by expanded trinucleotide repeats that lead, by a cascade of events, to the aberrant splicing of a large number of genes. These splicing variations are the cause of most of the disease symptoms. The authors use a reporter of one of those downstream splicing events as an indicator of disease and response to treatment. To do so, they have generated a transgenic mouse model with a mutually exclusive double fluorescence reporter system: when exon 22 of the ATP2A1 gene is included (as seen in WT muscle) the DsRed reporter should be expressed, while when exon 22 is spliced out, as seen in DM muscle or regenerating fibres, GFP is expressed instead.

To test this method, they first expressed the reporter plasmid in a WT background and evaluated the response to injury. A time point analysis shows how GFP protein is highly expressed while regeneration is happening, while the red/green ratio returns to normal after this process is completed. When crossed with a mouse model for DM1 (creating a bitransgenic mouse), the animals showed a reduced DsRed/GFP ratio that could be restored when treatments were administered to the mice.

Finally, they evaluate the convenience of a spectroscopy method as opposed to a fluorescent microscopy method and also use it to evaluate new conjugated antisense oligonucleotide drugs that should offer enhanced distribution from the standard ones.

The manuscript is very well written and presents a comprehensive set of experiments to fully characterise the new model they propose. As a tool for the development of new drugs that alter slicing events in MD1 could be a very welcomed addition to other methods.

I only have a few minor comments to make:

- Please indicate those experiments in which administration was subcutaneous or intramuscular, as the biodistribution may be affected by it.
- I find very interesting the different results between different muscles studied. Why did the authors select those particular muscles, and have they analysed any other ones?
- Paragraph starting on line 278 and corresponding figures: I find it confusing to use ASO 948 and ASO 236 when only a few lines above they were referred as LICA ASO 992948 and ASO 445236 (if I did understand correctly). It might be more clear to be referred as LICA ASO vs. ASO, or “conjugated vs unconjugated “ ASO.
- I am not qualified to comment on the technical details of the spectroscopy vs fluorescent methodologies, but I think perhaps an important point the authors did not make is that the latter could be more prone to bias, as ROIs are selected by the operator.

Reviewers' comments

Reviewer #1 (Remarks to the Author):

This manuscript submitted to Nature Communications reports the development of “therapy reporter” (TR) mice model and fluorescence spectroscopy as drug development tools for evaluation of therapeutic oligonucleotide targeting of skeletal muscle.

1. The technology developed is interesting. But it seems the technique reported is limited for evaluation of drugs only for targeting of muscle tissue.

Response: By design, expression of the TR transgene is restricted to skeletal muscle due to the Human Skeletal Actin promoter/enhancer (Fig. 1a) (Crawford, et al., 2000). Prevention of transgene expression in skin, blood vessels, and connective tissue enhances sensitivity to fluorescence protein measurements in muscle tissue, which is ideal for screening drug candidates to treat muscle diseases. Replacement of the promoter would enable evaluation of drugs targeting non-muscle tissues.

Reference:

- Crawford, G.E. et al. Assembly of the dystrophin-associated protein complex does not require the dystrophin COOH-terminal domain. *J Cell Biol* **150**, 1399-1410 (2000).

2. Furthermore, the absorption and emission wavelengths of DeRed and GFP are very different and also not in the NIR-window, and both reporters suffer limited tissue penetration capability. These factors make it is very challenge to do the quantification analysis of the imaging signal.

Response: We agree that the emission spectra from GFP and DsRed are more vulnerable to distortion resulting from propagation through wavelength-dependent tissue optical properties than are fluorophores emitting in the NIR. This is precisely the rationale for the incorporation of white-light reflectance in the spectroscopy system that we designed and built for this study (Fig. 3). Acquisition of a broadband white-light spectrum characterizes the wavelength-dependent attenuation of the tissue. Wu et al., demonstrated very nicely that division of the measured

fluorescence by this white-light reflectance corrects the fluorescence for distortions introduced by the tissue optical properties. Co-authors T.M.B., S.M. and T.H.F. of this manuscript also have used this method successfully in several prior studies (**Baran**, et al., 2010; **Baran and Foster** 2011; Cottrell, et al., 2008; Finlay, **Mitra**, and **Foster**, 2002; Lee, Baron, and **Foster**, 2008). With respect to spectral analysis using singular value decomposition, relative to previous tissue spectroscopy challenges that we have addressed (Finlay, et al., 2001), the separation of GFP, DsRed, and endogenous tissue fluorescence represents a relatively straightforward problem.

In mice, histological measurements of epidermis, dermis, and hypodermis combine for approximately 350 μm in females and 450 μm in males (Calabro, et al., 2011). After hair removal, the skin in TR mice is nearly translucent, with the gastrocnemius and lumbar paraspinal muscles easily visible immediately beneath it. Even with limited tissue penetration capability, DsRed and GFP fluorescence is sufficiently bright to enable visualization of superficial muscle tissue and overlying blood vessels with high spatial resolution by fluorescence microscopy (Supplementary Fig. 5) or the human eye looking through the microscope binocular eyepiece. In the revised manuscript, we have added new data (Fig. 8) demonstrating that fluorescence spectroscopy measurements show strong correlation with alternative splicing by RT-PCR (also see Point 3 below).

References:

- **Baran, T.M.**, Giesselman, B.R., Hu, R., Biel, M.A. & **Foster, T.H.** Factors influencing tumor response to photodynamic therapy sensitized by intratumor administration of methylene blue. *Lasers Surg Med* **42**, 728-735 (2010).
- **Baran, T.M.** & **Foster, T.H.** New Monte Carlo model of cylindrical diffusing fibers illustrates axially heterogeneous fluorescence detection: simulation and experimental validation. *J Biomed Opt* **16**, 085003 (2011).
- Calabro, K., Curtis, A., Galarneau, J.R., Krucker, T. & Bigio, I.J. Gender variations in the optical properties of skin in murine animal models. *J Biomed Opt* **16**, 011008 (2011).

- Cottrell, W.J., Paquette, A.D., Keymel, K.R., **Foster, T.H.** & Oseroff, A.R. Irradiance-dependent photobleaching and pain in delta-aminolevulinic acid-photodynamic therapy of superficial basal cell carcinomas. *Clin Cancer Res* **14**, 4475-4483 (2008).
- Finlay, J.C., Conover, D.L., Hull, E.L. & **Foster, T.H.** Porphyrin bleaching and PDT-induced spectral changes are irradiance dependent in ALA-sensitized normal rat skin in vivo. *Photochem Photobiol* **73**, 54-63 (2001).
- Finlay, J.C., **Mitra, S.** & **Foster, T.H.** In vivo mTHPC photobleaching in normal rat skin exhibits unique irradiance-dependent features. *Photochem Photobiol* **75**, 282-288 (2002).
- Lee, T.K., Baron, E.D. & **Foster, T.H.** Monitoring Pc 4 photodynamic therapy in clinical trials of cutaneous T-cell lymphoma using noninvasive spectroscopy. *J Biomed Opt* **13**, 030507 (2008).
- Wu, J., Feld, M.S. & Rava, R.P. Analytical model for extracting intrinsic fluorescence in turbid media. *Appl Opt* **32**, 3585-3595 (1993).

3. The reliability of using DsRed/GFP ratio needs more validation.

Response: We agree that it is important to validate the DsRed/GFP values, which are determined by splicing in the most superficial fibers of each muscle, as accurate estimates of alternative splicing in the entire muscle. Throughout the manuscript we demonstrate that low DsRed/GFP ratios determined by either imaging or spectroscopy are associated with low exon 22 inclusion of both the *ATP2A1* minigene and endogenous mouse *Atp2a1*, while high DsRed/GFP ratios are associated with high exon 22 inclusion of each. To highlight this point further, in the revised manuscript we have added new graphs (Fig. 8) that demonstrate strong correlation of spectroscopy measurements of DsRed/GFP fluorescence in gastrocnemius and paraspinal muscles with splicing of the *ATP2A1* transgene and endogenous *Atp2a1* in these muscles, validating that fluorescence measurements in the most superficial muscle fibers are accurate estimates of splicing outcomes in the entire muscle.

Reviewer #2 (Remarks to the Author):

In this paper Hu and colleagues have generated a novel reporter bi-transgenic mouse model of DM1 based on the well-characterized and muscle-restricted HSALR mouse model. They

customized a bi-chromatic reporter system in which the expression of either DsRed or GFP is determined by splicing of an upstream minigene containing ATP2A1 exon 22 that is misregulated in DM1. Moreover, they validated a non-invasive *in vivo* fluorescence spectroscopy system to monitor splicing changes and evaluate therapeutic activity of ASOs, including a new LICA-ASO.

The manuscript is clear and well written. Experimental evidences support the use of the bi-transgenic mouse model (TR:HSALR) in association with *in vivo* fluorescence spectroscopy as drug development tools for DM1. These combined non-invasive tools will be useful to evaluate muscle-specific efficacy of therapeutic approaches for DM1 using a primary screening based on ATP2A1 exon 22 missplicing correction. However, this reporter model cannot assess effects of potential therapies on other molecular alterations that were also described in DM1 muscles as abnormal activation of CELF1 or GSK3beta. Nevertheless, the proposed tools provide a novel resource for translational research in DM1.

Comments and concerns:

1. Because ATP2A1 exon 22 contains stop codons when expressed in frame, authors should clearly explain that the ATP2A1 minigene has been modified to enable expression of DsRed when ATP2A1 exon 22 is included.

Response: In the Methods section, we explain the modification of exon 22 as follows: “To enable shift of the reading frame, exon 22 of the *ATP2A1* minigene was modified by site-directed mutagenesis to induce a single base pair deletion so that it contains 41 base pairs instead of 42.”

2. Plasmid experiments (Fig S1c-g) showed that DsRed is decreased in a DM1 context. However it is not clear whether a concomitant increase of GFP is measured!

Response: In Supplementary Fig. 1c - g, GFP expression is higher in HSA^{LR} muscles injected with the plasmid than in WT muscles injected with an equal amount of plasmid. This is most evident by RT-PCR in Supplementary Fig. 1e (lower exon 22 inclusion percentage) and by quantitative fluorescence microscopy of muscle tissue sections shown in Supplementary Fig. 1g. For the *in vivo* fluorescence images in Supplementary Fig. 1c, we set the intensity scale at a

maximum of 2600 to show differential DsRed expression in WT and HSA^{LR} muscle. Lowering the intensity scale further in order to demonstrate the GFP expression would give the impression that both of the DsRed images are saturated (neither are).

3. An unexpected exclusive nuclear localization of GFP was observed whereas DsRed is cytoplasmic. This point should be addressed.

Response: The original RG6 construct (Orengo, et al., 2006) contains a nuclear localizing signal, and expression of both DsRed and GFP was restricted to nuclei. During the cloning process, we removed the nuclear localizing signal, expecting that DsRed and GFP both would be localized to the cytoplasm. However, even in the absence of the nuclear localizing signal, expression of GFP expression by the TR construct was restricted to myonuclei, suggesting the possibility that the N-terminal peptide sequence upstream of GFP in the TR construct may contain a nuclear localizing signal. We have added this point to the Discussion (paragraph 2).

Reference:

- Orengo, J.P., Bundman, D. & Cooper, T.A. A bichromatic fluorescent reporter for cell-based screens of alternative splicing. *Nucleic Acids Res* **34**, e148 (2006)

4. Expression of GFP in a DM1 context was confirmed by Western blot in Fig 2e. What about the quantification of DsRed expression in WT and DM1 condition?

Response: We were unable to find an antibody that recognizes DsRed protein by immunoblot or immunofluorescence. However, in Fig. 2f we use quantitative fluorescence microscopy of muscle tissue sections to demonstrate that DsRed fluorescence is brighter in TR single transgenic (WT condition) than in TR;HSA^{LR} bi-transgenic (DM1 condition) mice.

5. Data in Fig 6 and 7 were generated from 2 mice each group only. How appropriate are the proposed statistical analyses? Authors should include either additional mice or another independent experiment to strength their results and conclusion.

Response: We discuss sample size in the Methods section. Before our first experiment, the response to ASO treatment in the TR;HSA^{LR} mice as measured by *in vivo* imaging or spectroscopy was unknown. Therefore, we were unable to choose a sample size ahead of time to ensure adequate power to measure pharmacodynamic activity. Instead we estimated sample sizes based on RT-PCR splicing patterns of TR-ATP2A1 exon 22 and *in vivo* spectroscopy measurements of DsRed/GFP values in gastrocnemius muscles of TR single transgenic and TR;HSA^{LR} bi-transgenic mice. Using RT-PCR, the difference between means of these two groups was 31 and the standard deviations were 1.6 and 1.7, respectively, meaning that a sample size of 1 from each group would provide 98% power to detect a difference with $P < 0.05$. Using spectroscopy measurements, the difference between means of these two groups was 4.2 and the standard deviations 0.02 and 1.0, respectively, indicating that a sample size of 2 from each group (N = 4 total) would provide 98% power to detect a difference with $P < 0.05$.

In addition to the data shown in Figs. 6 and 7 of the first draft, we did two earlier experiments testing saline vs. LICA oligo 992948 at a dose of 25 mg/kg twice per week for four weeks. We have added these data to the revised manuscript as Supplementary Fig. 6. Both experiments showed that the LICA oligo had a robust effect by *in vivo* fluorescence microscopy or spectroscopy, alternative splicing of transgene *ATP2A1* and endogenous mouse *Atp2a1*, and target knockdown by ddPCR. Based on these results, we did two dose response experiments, one testing three different doses of the LICA oligo vs. saline (Fig. 6), and a second testing two different doses of the LICA oligo vs. identical doses of the unconjugated parent ASO (Supplementary Fig. 8).

In this revision, we have added new data from a fifth experiment testing the LICA oligo, which also is the second experiment head-to-head against the unconjugated parent ASO (N = 3 each). The results from this experiment confirm that mice treated with the LICA oligo demonstrate an earlier and more rapid increase in DsRed/GFP values in both gastrocnemius and lumbar paraspinal muscles, a greater knockdown of target *ACTA1* transcripts by ddPCR, and more complete splicing correction by RT-PCR in gastrocnemius and lumbar paraspinal muscles than mice treated with the unconjugated ASO (Fig. 7). Stopping the most recent experiment two weeks earlier than the previous head-to-head experiment (Day 28 rather than Day 42) may have helped to highlight these differences.

6. Typo error in Fig S6a-b: doses in panel “a” (2.5, 8.5 and 25mg/kg) are different from those in panel “b” (8.5, 12.5 and 25mg/kg)!

Response: Thank you for catching this error. It has been corrected.

Reviewer #3 (Remarks to the Author):

This manuscript describes the development and characterisation of a new in vivo model to test potential therapies for myotonic dystrophy. Myotonic dystrophy is caused by expanded trinucleotide repeats that lead, by a cascade of events, to the aberrant splicing of a large number of genes. These splicing variations are the cause of most of the disease symptoms. The authors use a reporter of one of those downstream splicing events as an indicator of disease and response to treatment. To do so, they have generated a transgenic mouse model with a mutually exclusive double fluorescence reporter system: when exon 22 of the ATP2A1 gene is included (as seen in WT muscle) the DsRed reporter should be expressed, while when exon 22 is spliced out, as seen in DM muscle or regenerating fibres, GFP is expressed instead. To test this method, they first expressed the reporter plasmid in a WT background and evaluated the response to injury. A time point analysis shows how GFP protein is highly expressed while regeneration is happening, while the red/green ratio returns to normal after this process is completed. When crossed with a mouse model for DM1 (creating a bitransgenic mouse), the animals showed a reduced DsRed/GFP ratio that could be restored when treatments were administered to the mice. Finally, they evaluate the convenience of a spectroscopy method as opposed to a fluorescent microscopy method and also use it to evaluate new conjugated antisense oligonucleotide drugs that should offer enhanced distribution from the standard ones. The manuscript is very well written and presents a comprehensive set of experiments to fully characterise the new model they propose. As a tool for the development of new drugs that alter splicing events in MD1 could be a very welcomed addition to other methods.

I only have a few minor comments to make:

1. Please indicate those experiments in which administration was subcutaneous or intramuscular, as the biodistribution may be affected by it.

Response: We agree this is an important point, and have clarified that the experiments described in Figs. 3 and 4, and Supplementary Figs. 1 and 4 involved direct intramuscular injections, and that the experiments described in Figs. 5 - 9, and Supplementary Figs. 5 - 8 involved subcutaneous injections.

2. I find very interesting the different results between different muscles studied. Why did the authors select those particular muscles, and have they analysed any other ones?

Response: The gastrocnemius and lumbar paraspinal muscles are large and well positioned directly beneath the skin for easy imaging in the prone position (see Fig. 4b). By contrast, the quadriceps muscle, which is frequently studied in models of muscular dystrophies, is deeper beneath the skin and partially covered by a layer of adipose tissue, which obscures non-invasive fluorescence measurements. A second muscle that is studied frequently, the tibialis anterior (TA), is accessible for imaging but demonstrates lower fluorescence expression in these mice than the gastrocnemius or lumbar paraspinal muscles and, therefore detection of drug target engagement by fluorescence measurements in the TA is less robust.

3. Paragraph starting on line 278 and corresponding figures: I find it confusing to use ASO 948 and ASO 236 when only a few lines above they were referred as LICA ASO 992948 and ASO 445236 (if I did understand correctly). It might be more clear to be referred as LICA ASO vs. ASO, or “conjugated vs unconjugated “ ASO.

Response: We agree it was confusing. To improve clarity, we now use the terms “LICA” for the conjugated ASO, and “ASO” for the unconjugated ASO.

4. I am not qualified to comment on the technical details of the spectroscopy vs fluorescent methodologies, but I think perhaps an important point the authors did

not make is that the latter could be more prone to bias, as ROIs are selected by the operator.

Response: This is a good point, and we agree that elimination of operator selected ROIs is an additional advantage of spectroscopy measurements.

Reviewer #1 (Remarks to the Author):

The revised manuscript address the reviewer's concerns and the paper is acceptable for publication.

Reviewer #2 (Remarks to the Author):

Overall, comments/concerns were taken into consideration (see minor points below). However in the new Figure 7 (that was added in the revised version), statistical analyses performed between the three groups (Saline, ASO and LICA) in panels c and e are not appropriate (N=1 in the control/Saline group). Either the number of mice in the control group (Saline) should be increased (N=3) for multi-group comparison (ANOVA) or a two-group comparison (t-test) should be performed to compare ASO to LICA only (a previous study has already compare ASO to Saline).

Minor points:

-1. Because ATP2A1 exon 22 contains stop codons when expressed in frame, authors should clearly explain that the ATP2A1 minigene has been modified to enable expression of DsRed when ATP2A1 exon 22 is included.

Response: In the Methods section, we explain the modification of exon 22 as follows: "To enable shift of the reading frame, exon 22 of the ATP2A1 minigene was modified by site- directed mutagenesis to induce a single base pair deletion so that it contains 41 base pairs instead of 42."

Re: The fact that the exon 22 was modified should also be specified in the main text (page 6).

- 2. Plasmid experiments (Fig S1c-g) showed that DsRed is decreased in a DM1 context. However it is not clear whether a concomitant increase of GFP is measured!

Response: In Supplementary Fig. 1c - g, GFP expression is higher in HSALR muscles injected with the plasmid than in WT muscles injected with an equal amount of plasmid. This is most evident by RT-PCR in Supplementary Fig. 1e (lower exon 22 inclusion percentage) and by quantitative fluorescence microscopy of muscle tissue sections shown in Supplementary Fig. 1g. For the in vivo fluorescence images in Supplementary Fig. 1c, we set the intensity scale at a maximum of 2600 to show differential DsRed expression in WT and HSALR muscle. Lowering the intensity scale further in order

to demonstrate the GFP expression would give the impression that both of the DsRed images are saturated (neither are).

Re: The last argument is justified but GFP images (without DsRed) would allow visualizing the increase.

Reviewer #3 (Remarks to the Author):

The authors have addressed my comments in a satisfying manner and I support the publication of this manuscript.

Best regards

Reviewers' comments:

Reviewer #1 (Remarks to the Author):

The revised manuscript address the reviewer's concerns and the paper is acceptable for publication.

Reviewer #2 (Remarks to the Author):

Overall, comments/concerns were taken into consideration (see minor points below). However in the new Figure 7 (that was added in the revised version), statistical analyses performed between the three groups (Saline, ASO and LICA) in panels c and e are not appropriate (N=1 in the control/Saline group). Either the number of mice in the control group (Saline) should be increased (N=3) for multi-group comparison (ANOVA) or a two-group comparison (t-test) should be performed to compare ASO to LICA only (a previous study has already compare ASO to Saline).□

Response: In Fig. 7, we used two-way ANOVA to compare the response of alternative splicing to two different drugs in four different muscles, for a total of four comparison groups (N = 3 each group), which is more appropriate than using four separate t-tests. We removed the saline comparison data from the statistical analysis, and have updated the graphs in Fig. 7 panels c and e accordingly.

Minor points:

-1. Because ATP2A1 exon 22 contains stop codons when expressed in frame, authors should clearly explain that the ATP2A1 minigene has been modified to enable expression of DsRed when ATP2A1 exon 22 is included. □**Response:** In the Methods section, we explain the modification of exon 22 as follows:□“To enable shift of the reading frame, exon 22 of the ATP2A1 minigene was modified by site- directed mutagenesis to induce a single base pair deletion so that it contains 41 base pairs instead of 42.” □□

The fact that the exon 22 was modified should also be specified in the main text (page 6).

Response: We agree that this is an important point, and have added the following sentence to the page 6 of the manuscript main text, “To bypass a stop codon and enable shift of the reading

frame (Kimura, et al., 2005), exon 22 of the *ATP2A1* minigene was modified by site-directed mutagenesis to induce a single base pair deletion so that it contains 41 base pairs instead of 42.” We also have included this sentence in the Methods section (manuscript page 19).

Reference:

- Kimura, T. et al. Altered mRNA splicing of the skeletal muscle ryanodine receptor and sarcoplasmic/endoplasmic reticulum Ca²⁺-ATPase in myotonic dystrophy type 1. *Hum Mol Genet* **14**, 2189-2200 (2005).

- 2. Plasmid experiments (Fig S1c-g) showed that DsRed is decreased in a DM1 context. However it is not clear whether a concomitant increase of GFP is measured! □Response: In Supplementary Fig. 1c - g, GFP expression is higher in HSALR muscles injected with the plasmid than in WT muscles injected with an equal amount of plasmid. This is most evident by RT-PCR in Supplementary Fig. 1e (lower exon 22 inclusion percentage) and by quantitative fluorescence microscopy of muscle tissue sections shown in Supplementary Fig. 1g. For the *in vivo* fluorescence images in Supplementary Fig. 1c, we set the intensity scale at a maximum of 2600 to show differential DsRed expression in WT and HSALR muscle. Lowering the intensity scale further in order to demonstrate the GFP expression would give the impression that both of the DsRed images are saturated (neither are). □□

The last argument is justified but GFP images (without DsRed) would allow visualizing the increase. □□□□

Response: The raw GFP quantitation by *in vivo* imaging in this experiment was about the same, while the DsRed quantitation was substantially higher in the WT than in the HSA^{LR} muscle. The ratio of DsRed/GFP is the key measurement.

Reviewer #3 (Remarks to the Author):

The authors have addressed my comments in a satisfying manner and I support the publication of this manuscript. □□Best regards.